# Zero-preserving imputation of single-cell RNA-seq data

George C. Linderman [1,8], Jun Zhao[2,8], Manolis Roulis [3], Piotr Bielecki [3,7], Richard A. Flavell [3,4], Boaz Nadler [5] & Yuval Kluger [1,2,6✉]

A key challenge in analyzing single cell RNA-sequencing data is the large number of false zeros, where genes actually expressed in a given cell are incorrectly measured as unexpressed. We present a method based on low-rank matrix approximation which imputes these values while preserving biologically non-expressed genes (true biological zeros) at zero expression levels. We provide theoretical justification for this denoising approach and demonstrate its advantages relative to other methods on simulated and biological datasets.

[1] Program in Applied Mathematics, Yale University, New Haven, CT 06511, USA. [2] Interdepartmental Program of Computational Biology and Bioinformatics, Yale University, New Haven, CT 06511, USA. [3] Department of Immunobiology, Yale University, New Haven, CT 06511, USA. [4] Howard Hughes Medical Institute, Yale University School of Medicine, New Haven, CT, USA. [5] Department of Computer Science and Applied Mathematics, Weizmann Institute of Science, Rehovot, Israel. [6] Department of Pathology, Yale University, New Haven, CT 06511, USA. [7] Present address: Celsius Therapeutics, Cambridge, USA. [8] These authors contributed equally: George C. Linderman, Jun Zhao. ✉email: yuval.kluger@yale.edu

Single-cell RNA-sequencing (scRNA-seq) techniques measure gene expression at the individual cell level. This requires amplification of truly minute quantities of mRNA, resulting in a phenomenon called "dropout" in which an expressed transcript is not detected and hence assigned a zero expression value. As a result, we define two types of zeros in the observed matrix: biological zeros (genes that were not expressed at the time of sequencing) and technical zeros (genes that were expressed at the time of sequencing but not measured). We note that our definition of "technical zeros" includes "sampling zeros," which are the result of undersampling the transcripts in each cell[1,2]. Roughly 15–40% of genes in bulk RNA-seq of different tissues are not expressed[3], suggesting that at least that percentage of genes are not expressed in individual cells and are thus biological zeros. However, due to technical zeros and cell-to-cell heterogeneity, the fraction of zeros in a typical scRNA-seq expression matrix is much higher, even exceeding 90% with recently developed droplet-based techniques.

Due to a large number of technical zeros, directly processing the raw data may be detrimental to downstream analysis, such as clustering and visualization. Imputation of the missing values has the potential to improve these analyses. An imputation method for scRNA-seq data should have (at least) the following two properties: (i) it should accurately impute the data, in particular preserving at zero true biological zeros while completing the technical ones. We illustrate this point below using a dataset from[4], whereby detecting which matrix entries are true biological zeros, has crucial importance for a biological understanding of the particular system of interest. (ii) it should be able to process, in a reasonable time, large-scale scRNA-seq datasets that contain hundreds of thousands to millions of cells.

Over the past few years, several imputation methods have been developed, including DCA[5], MAGIC[6], SAVER[7], and scImpute[8]. Some methods such as DCA and MAGIC treat all zeros as "missing data" and output a matrix in which every gene is expressed in every cell (i.e., there are no biological zeros). While other methods such as SAVER and scImpute do preserve biological zeros, as we show below, they either detect relatively few of them or otherwise impute a few of the technical ones. Finally, with their current implementations, SAVER and scImpute may be extremely slow on scRNA-seq datasets with over 100,000 cells.

To address these challenges, we present Adaptively thresholded Low-Rank Approximation (ALRA), a method for imputation of scRNA-seq data. ALRA takes advantage of the non-negativity and low-rank structure of an expression matrix to selectively impute technical zeros (Fig. 1). Due to widespread correlations between genes across different cells, we assume that the true expression matrix is non-negative, low-rank, and contains many zeros (biological zeros). The matrix measured in a scRNA-seq experiment is a corrupted version of this expression matrix where many technical zeros are introduced by measurement error or insufficient total read counts. The first step of ALRA is to compute a low-rank approximation of the observed matrix using the singular vector decomposition (SVD). The rank $k$ of the approximation is automatically determined by a simple procedure described in the Online Methods. The matrix resulting from this low-rank approximation, in general, has no zeros. Hence, the last step of ALRA is to restore the biological zeros in the matrix by thresholding its entries. As we prove mathematically in the supplement, under certain assumptions, the entries of this matrix that correspond to the true biological zeros are symmetrically distributed around zero. Due to this symmetry, the negative values provide an estimate for the error distribution of the elements corresponding to true biological zeros. As such, we set to zero all the entries of each row (or gene) that are smaller than the magnitude of the most negative value (see Methods for details). This ensures that with high probability all of the entries which correspond to biological zeros are set to zero. Our analysis may be of independent interest to other low-rank matrix completion or denoising problems.

We emphasize the simplicity—and hence computational efficiency—of our approach: it is SVD followed by a thresholding scheme that takes advantage of the non-negativity of the true matrix. Remarkably, as we demonstrate below with experiments on 11 real scRNA-seq datasets, this simple approach outperforms other more complex methods for the recovery of scRNA-seq expression data. In addition to these empirical results, we also give a rigorous theoretical justification for ALRA's algorithm (see supplement).

## Results and discussion

**ALRA preserves biological zeros**. Using a variety of datasets where ground truth is known, we empirically illustrate ALRA's ability to preserve biological zeros. We first considered scRNA-seq of purified populations of peripheral blood mononuclear cells (PBMCs) generated by Zheng et al. (2017)[9]. We merged the purified populations into a single matrix and ran ALRA, DCA, MAGIC, SAVER, and scImpute on the merged matrix. We then focused our attention on B cells, natural killer cells, cytotoxic T cells, and helper T cells. These four cell types are well characterized and there are known marker genes specific to each. For example, *PAX5* (*BSAP*), *NCAM1* (*CD56*), *CD8A*, and *CD4* are marker genes for peripheral B cells, NK cells, cytotoxic T cells, and helper T cells, respectively[10–12]. The expression values of a marker gene in a population that is known to not express that marker gene (e.g., *NCAM1* in B cells) are biological zeros. Note that there may be further biological zeros in the marker genes due to cell-to-cell heterogeneity (e.g., some CD8+ T cells may not

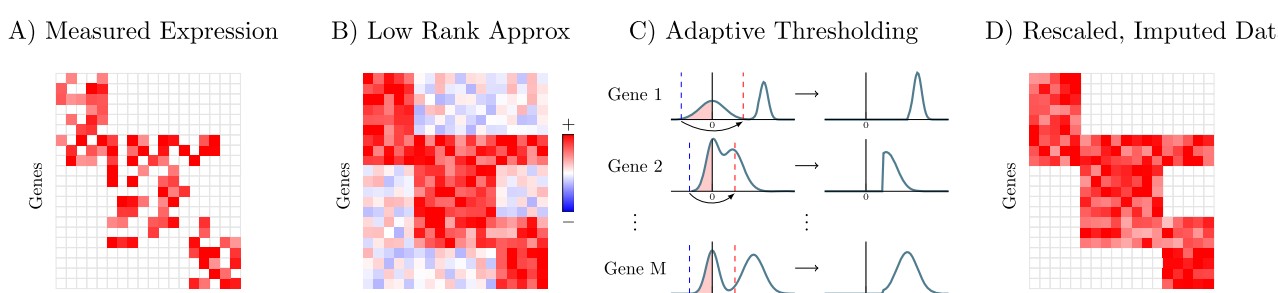

**Fig. 1 Overview of the ALRA imputation scheme. A** Measured expression matrix contains technical zeros (in each block) and biological zeros (outside each block). **B** Low-rank approximation via SVD results in a matrix where every zero is completed with a non-zero value (i.e., biological zeros are not preserved). **C** The elements corresponding to biological zeros for each gene are symmetrically distributed around 0, so the magnitude of the most negative elements in each row is also approximately the magnitude of the most positive values assigned to biological zeros. Therefore, thresholding the values in each row, restores almost all of the biological zeros in the imputed matrix (**D**).

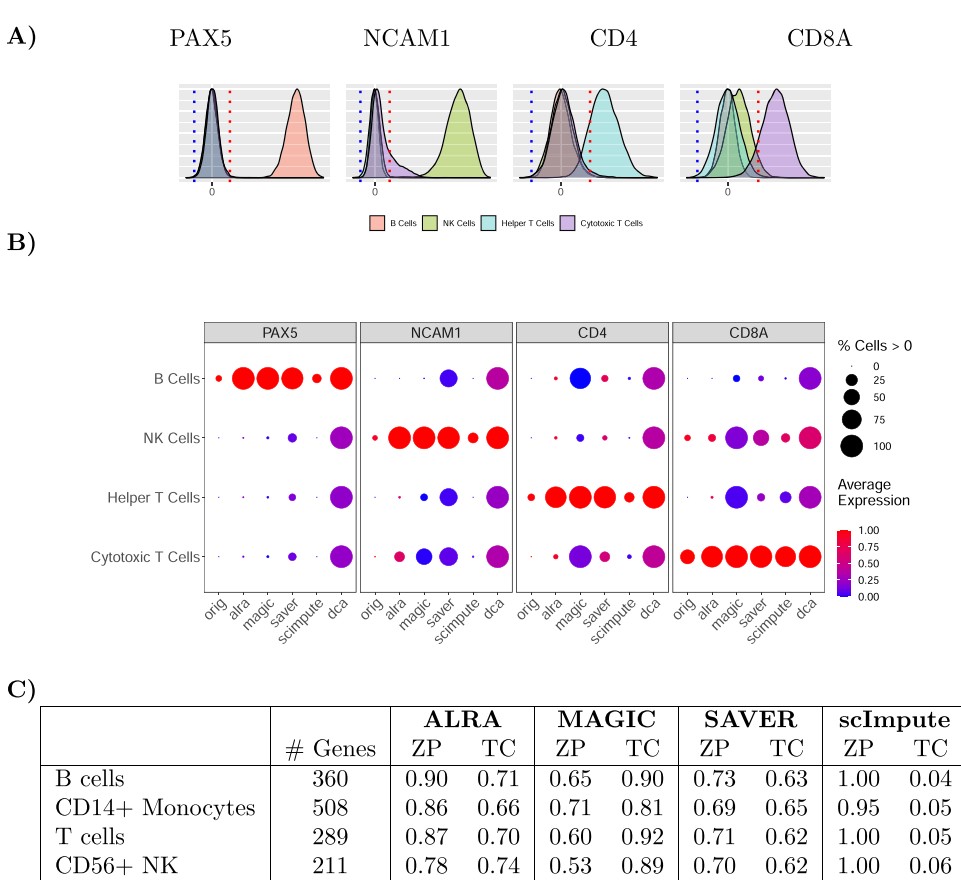

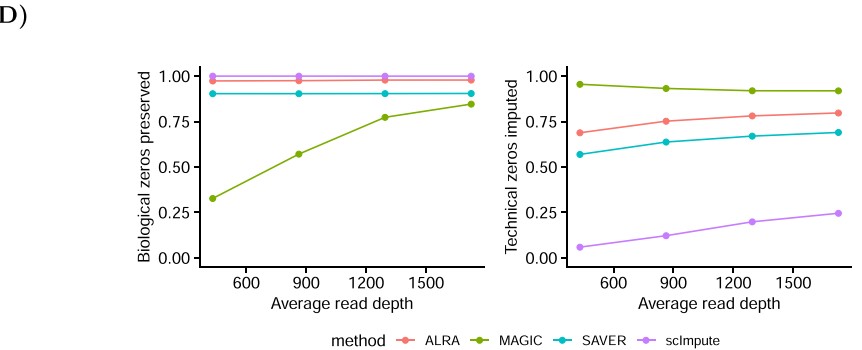

**Fig. 2 Preserving biological zeros in purified PBMC populations. A** After low-rank approximation, ALRA recovers biological zeros of each gene by thresholding (red line) the expression values of that gene in all cells at the magnitude of the $p = 0.001$ quantile of that gene (blue line). **B** Preserving biological zeros in genes known to be specific to each cell population. **C** Biological zeros were defined as genes which should not be expressed in a cell type (based on bulk RNA-seq). The ratio of these biological zeros preserved (ZP) is shown alongside the ratio of total zeros completed (TC). **D** Percentage of biological zeros preserved (left) and technical zeros completed (right) in a simulated scRNA-seq dataset as sequencing depth increases. DCA not shown for (**C**) and (**D**), as it imputes every zero to a positive number.

actually express *CD8* at the time of sequencing), and for this reason, we focus on the known biological zeros: genes for which there is solid biological knowledge that these are not expressed in a cell population.

After low-rank approximation, all entries in the matrix are non-zero. However, as shown in Fig. 2A and in accordance with our theoretical analysis, the elements of the matrix correspond to known biological zeros (e.g., the gene *PAX5* in NK cells, cytotoxic T cells, or helper T cells) are symmetrically distributed around zero. Thus in the second step of ALRA we threshold at the red dotted lines, which are the symmetrical mirror image of the 0.001 quantiles of each gene (blue dotted lines). As shown in Fig. 2B, this step recovers the known biological zeros of the

marker genes for these four purified peripheral blood mononuclear cell populations. We note that some cytotoxic T cells with NK cell markers (i.e., CD8+CD56+ cells) are known to be present in the peripheral blood, which justifies the completion of *NCAM1* in a small fraction of cytotoxic T cells[13].

In contrast, DCA, MAGIC, and SAVER output non-zero values for many entries in the matrix that are expected to be biological zeros. For example, both DCA and MAGIC result in a matrix where CD8+ T cells express CD4 at nontrivial levels. While there may be some crossover between these marker genes, particularly during development, it would be in a small fraction of the mature cells. Furthermore, the shapes of the distributions obtained by these three imputation methods have no obvious

cut points for thresholding the biological zeros, especially because in many cases the values that should be zero are not small (Supplementary Fig. 1). We also note that while scImpute preserves biological zeros, it only imputes 4–6% of the total zeros, hence clearly underperforming in the task of imputing technical zeros.

Next, going beyond the above illustrative example with only four genes, we demonstrate the ability of ALRA to preserve biological zeros in hundred of genes identified via bulk RNA-seq of purified immune cell populations. Specifically, genes that were not expressed in peripheral B cells, CD14+ monocytes, T cells, and CD56+ NK cells in bulk RNA-seq of Hoek et al. (2015)[14] were identified as "biological zeros." We evaluated the ratio of these biological zeros preserved (ZP) by the various imputation methods. ALRA preserved more than 85% of the zeros in B cells, CD14 monocytes, and T cells, and preserved 75% of the zeros in CD56 NK cells. DCA does not preserve any biological zeros (it always outputs values greater than zero), and hence is not shown in the table. MAGIC preserved between 53 and 71% of the biological zeros. scImpute preserved the most biological zeros, at the cost of hardly imputing any of the zero entries in the matrix (which includes a large number of technical zeros). SAVER imputes fewer total zeros than ALRA but only preserves 69–73% of the biological zeros, which is substantially lower than ALRA.

We performed a similar analysis using scRNA-seq data from CITE-seq of a peripheral immune cell population[15]. In CITE-seq data, both single-cell RNA-seq and cell surface protein expression are measured for each of the cells. Unlike with the previous experiment where FACS was used to separate the cells into cell types and then sequence them, in this dataset we can identify B cells, T cells, and monocytes based on the cell surface markers. We then ran the imputation methods on the scRNA-seq data and evaluated the ratio of biological zeros preserved by different imputation methods (Supplementary Table 2). As before, ALRA preserved substantially more biological zeros than MAGIC and SAVER. scImpute preserved 5-8% more zeros than ALRA, but only imputes about a fifth of the zeros that ALRA completes.

Finally, we assessed ALRA's ability to selectively complete technical zeros (while preserving biological zeros) using a simulated dataset where the truth is known. We used bulk RNA-seq of 5 T-cell populations from the ImmGen consortium[16] to simulate scRNA-seq expression data. For each population, we generated single-cell profiles by sampling counts from a multinomial distribution as in refs. [17,18] with probabilities proportional to expression in the bulk profile. Specifically, for a given population, the probability vector is equal to the corresponding bulk profile normalized to sum to one. We then sampled from each multinomial distribution with an increasing number of reads. The probability of recovering a technical zero as a function of the underlying expression is shown in Supplementary Fig. 11. As shown in Fig. 2D, ALRA preserves ~97% of the true biological zeros while completing a large proportion of the non-zeros, even at shallow read depths. In contrast, MAGIC preserves many fewer zeros, especially at shallow read depths. SAVER fares better, but still incorrectly completes more biological zeros than ALRA. Finally, scImpute preserves all the biological zeros, but only imputes less than 25% of the technical zeros. In these experiments, DCA was not included because it outputs a matrix of all positive values (i.e., with no zeros), and hence never preserves biological zeros.

**The importance of preserving biological zeros**. A cell type or cellular process is often characterized by both positive marker genes (genes expressed by the cells) and negative marker genes (genes not expressed by the cells). Imputation methods that do not preserve biological zeros result in a matrix where every gene is expressed by every cell, requiring the practitioner to manually threshold the expression values to determine which cells express or do not express the markers. This thresholding can be arbitrary and lead to erroneous results. In contrast, ALRA does not require this thresholding step, as cells not expressing the gene will not be falsely imputed to non-zero values. We demonstrate the importance of preserving biological zeros using three real-life examples.

Gupta et al.[4] recently found that Wnt signalling is necessary for hair follicle dermal condensate (DC) differentiation, an essential step for hair follicle development. This was established in mouse embryonic skin by conditionally knocking out beta-catenin, the key transducer of the Wnt pathway, and comparing the expression of DC marker genes with control using scRNA-seq. We imputed these expression values using ALRA and compared the ratio of cells expressing *Sox2* (a marker for differentiated DC cells) and the ratio of cells expressing *Axin2/Lef1* (markers for Wnt signaling) between the knockout and control to show that only dermal cells that retained Wnt activity were able to acquire DC cell fate. ALRA's ability to preserve the biological zeros allows us to quantitatively compare the number of cells expressing the marker genes (Supplementary Table 1). The ratio of the proportions of cells expressing *Sox2* and *Axin2/Lef1* between the knockout and control is similar to the ratio computed from the pre-imputed data. In particular, ALRA preserves the conclusion that Wnt signalling was conditionally knocked out in a proportion of cells within the mutants, and that there was a corresponding impairment in DC differentiation caused by the absence of Wnt signalling. In contrast, MAGIC and DCA both output a matrix with no zeros, and hence, this proportion cannot be evaluated (Supplementary Figs. 2 and 3). Furthermore, based on their density plots and t-SNEs, it is not clear how to threshold the imputed values to restore the zeros (Supplementary Figure 3). SAVER's results falsely suggest that the Wnt activity in the knockout and control is identical, whereas scImpute falsely suggests that DC differentiation is similar between knockout and control (Supplementary Table 1). In settings where lack of expression of a marker gene is used to characterize a population or has other biological importance, existing methods that do not preserve biological zeros are unsuitable.

Next, we demonstrate that ALRA allows practitioners to draw upon a rich literature of positive and negative markers for known cell types. In Supplementary Fig. 4, we demonstrate this application by identifying known subtypes of PBMCs from Zheng et al.[9] using well-known markers (closely based on the Human Blood Cell Atlas[19], see methods). This is not possible using the raw data, as very few (if any) cells would express all the positive markers. But existing imputation methods that do not preserve biological zeros would require an arbitrary thresholding step to determine if a cell is "positive" or "negative" for the given gene. Effectively, ALRA allows for automatic gating of cells based on well-known marker genes, which can be especially useful for distinguishing cell subtypes within a cluster (e.g., Naive B cells and Memory B cells).

In a third example, we apply ALRA to human bronchial epithelial cells from Lukassen et al.[20]. These authors sought to identify cells that expressed three SARS-CoV-2 entry proteins, *ACE2*, *TMPRSS2*, and *FURIN*. Their analysis was hampered by dropout, as only three cells were triple-positive. Using ALRA, we can identify 1309 of these triple-positive cells (Supplementary Fig. 5). When imputing with methods that do not preserve biological zeros, cells that are not actually triple-positive are imputed to express all three genes, hence requiring a thresholding step to set these incorrect values to zero. This thresholding is far from trivial, as different thresholds can lead to dramatically different results.

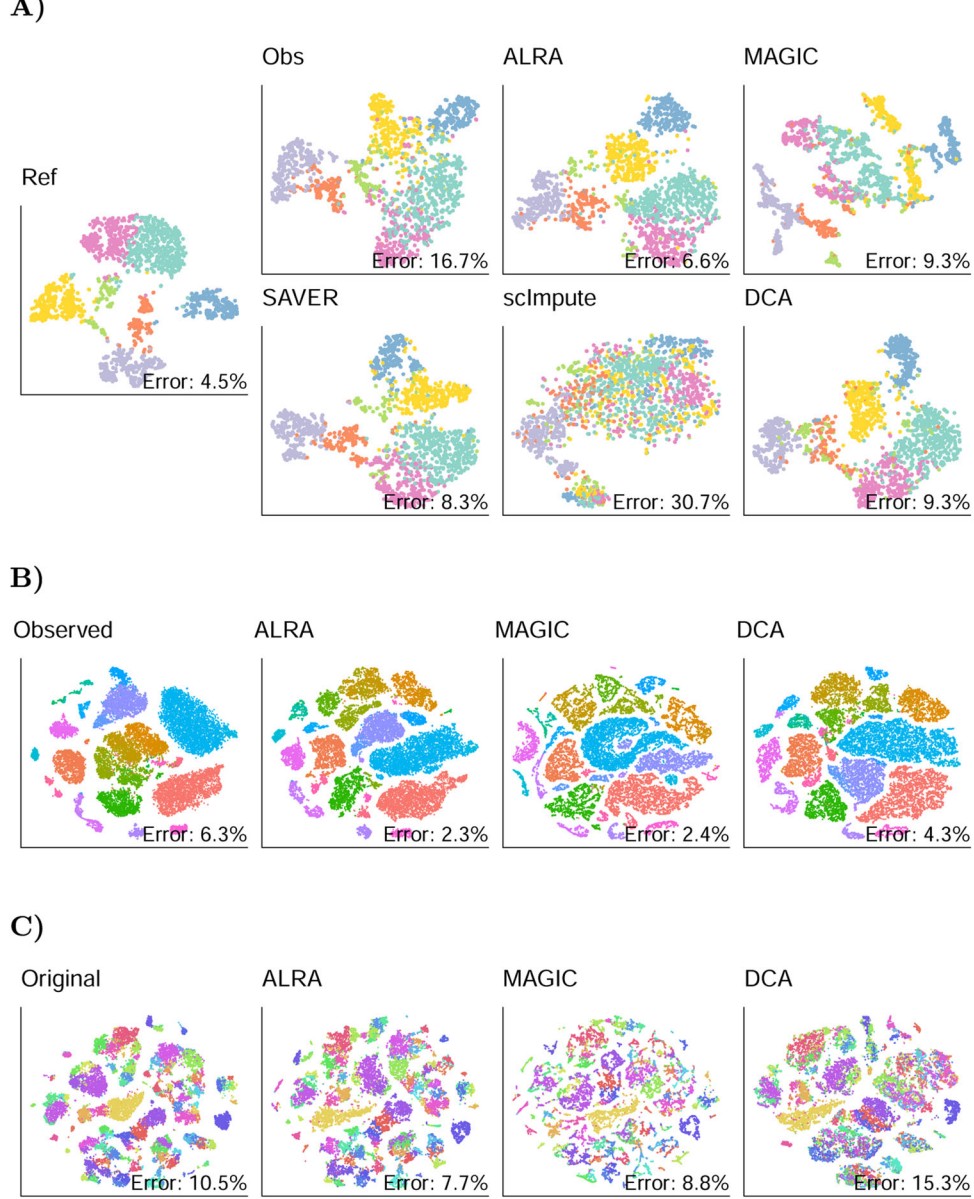

**Fig. 3 ALRA improves separation of cell types. A** t-SNE of mouse brain cells from Zeisel et al. were computed based on the original data (Ref). The reads were downsampled which introduced additional technical zeros and decreased the separation of different cell types (Obs). t-SNE after imputation by ALRA and other algorithms. To quantitatively compare separation of cell types, a random forest was trained to predict cell class based on expression values, and the out-of-bag classification error is shown in each plot. **B** The effect of imputation on separation of previously annotated cell types in mouse visual cortical cells from Hrvatin et al. and in mouse neocortical cells (**C**) from Tasic et al. Unlike in (**A**), the datasets were not downsampled prior to imputation as they already had high dropout rates.

**ALRA improves the separation of cell types**. We next evaluated ALRA's ability to improve the separation of clusters after synthetically introducing zeros. As in Huang et al., we downsampled the reads of Zeisel et al.[21], Baron et al.[22], Chen et al.[23], and La Manno et al.[24]. We showed that after imputation, cell types are more clearly separated in the t-SNE (Fig. 3A and Supplementary Figs. 6 to 8). To quantitatively confirm that ALRA increases separation, we trained supervised random forests to classify the cells into cell types based on expression values, and we compare the out-of-bag classification error before and after imputation. Imputation by ALRA substantially decreased the out-of-bag error on all four datasets, from average error of 16.4% to 8%. MAGIC, SAVER, and DCA also improved the average error to 10.3%, 9.8%, and 9.4%, respectively, whereas scImpute increased the error to 28%.

Additionally, we evaluate ALRA using t-SNE on two datasets with cells that the authors classified into distinct cell types. We applied ALRA, DCA, and MAGIC to the 65,539 mouse visual cortex cells from Hrvatin et al. (2018)[25] and 21,874 neocortical cells from Tasic et al. (2018)[26]. Figure 3 compares the t-SNE before and after imputation. After imputation by ALRA, several cell types in the Hrvatin et al. dataset are more clearly separated. While on the Tasic et al. dataset ALRA does not seem to significantly improve cell-type separation, at least it does not decrease separation, which DCA appears to do. We computed the out-of-bag classification error as before, and found that ALRA yielded an improved error from 6.3 to 2.3% in Hrvatin et al, and down from 10.5 to 7.7% in Tasic et al, confirming that ALRA improves the separation of the cell types. With their current implementations, scImpute and SAVER do not scale well to large

datasets and hence were not run on these matrices (see later for scalability benchmarks).

**Further validation of ALRA's imputation**. Having shown that ALRA successfully preserves biological zeros and improves the separation of cell types, we further validated ALRA performance with several additional analyses. First, we used the CITE-seq cord blood mononuclear cell (CBMC) dataset of Stoeckius et al. to show that after imputation, the RNA measurements of each gene are more consistent with the corresponding protein measurements (measured as antibody-derived tag (ADT)). We computed the $k$ nearest neighbors of each cell (for $k = 50, 100, 1000$) in the ADT space and also in the RNA space before and after imputation. ALRA nearly doubled the number of nearest neighbors that are consistent between the RNA and protein measurements (Supplementary Table 3). We also computed the pairwise distances between cells on the gene expression and on the protein expression data. We plotted the normalized distances between cells in gene expression data before and after imputation against the distances between cells in the protein expression data to show that ALRA improved their Spearman's correlation (Supplementary Fig. 9) from 0.36 to 0.64.

As in Huang et al., we further evaluated our approach by comparing ALRA's completion of Drop-seq data to RNA florescence in situ hybridization (FISH) of the same population of cells. In Torre et al. (2018), 8498 cells from a melanoma cell line were sequenced by Drop-seq[27]. RNA FISH was also used to measure the expression of 26 markers in cells of the same cell line. Although the gene expression profiles of individual cells cannot be compared across these two datasets, the distribution of gene expression should be consistent, because the cells were sampled from the same cell line. Huang et al. (2018) applied SAVER to the Drop-seq data and demonstrated that, as compared to the raw data, the estimated distribution of expression values is more consistent with the FISH distributions[7]. Their analysis computes the Gini coefficient of each gene, a measure of gene expression variability, which should be consistent across the Drop-seq and FISH technologies. Specifically, they showed that the Gini coefficients of a subset of marker genes in the FISH dataset were more consistent (correlation 0.6) with the SAVER estimates than either the raw Drop-seq data (correlation 0.3) or other completion methods (Supplementary Fig. 12). Our approach outperforms all other methods on this metric, obtaining a correlation of 0.8.

Next, we further evaluated ALRA using the downsampled datasets from before (Baron et al., Chen et al., La Manno et al., and Zeisel et al.). We computed the correlation of each gene/cell in the original dataset with the same gene/cell in the downsampled dataset before and after imputation by ALRA. We found that despite improving the clustering (see above), ALRA does not increase this correlation (Supplementary Fig. 15B). We attribute this to the presence of strong noise in the raw data which is not present after denoising using ALRA. To check if ALRA is preserving the signal, we computed the top $k$ PCs (where $k$ was chosen using the jackstraw method, see methods) of the original dataset with the top $k$ PCs of the downsampled dataset before and after imputation. We found that ALRA decreases the subspace angle between these signal PCs, from 0.34 to 0.25 for Baron et al, 0.30 to 0.17 for Chen et al, 0.36 to 0.24 for La Manno et al, and 0.21 to 0.15 for Zeisel et al. Thus ALRA effectively increases the correlation between the signals of the subsampled data and the original data. Additionally, the correlation of the imputed data with the original raw data (Supplementary Fig. 15B) is not a useful metric as it has a strong dependence on the noise of the original data.

We next focused on ALRA's ability to preserve subtle differences in expression values. A recent study uncovered the existence of two major lineages of fibroblasts in the intestine characterized by differential Pdgfra gene expression levels: $Pdgfra^{High}$ and $Pdgfra^{Low}$ fibroblasts[28]. We tested whether ALRA would preserve this expression pattern by performing an independent scRNA-seq experiment and sequencing fewer cells, thus generating a scarcer dataset to be completed by ALRA and other methods. We isolated mesenchymal/lamina propria cells from the colon of $Pdgfra^{EGFP/+}$ knock-in mice and used flow cytometry to confirm the presence of two major populations with non-zero Pdgfra-EGFP expression: $Pdgfra^{Low}$ cells and $Pdgfra^{High}$ cells (Supplementary Fig. 16). After imputation by ALRA, a Pdgfra expression pattern very similar to that observed by flow cytometry can be clearly seen. In contrast, other methods do not show two distinct levels of $Pdgfra$ expression (Supplementary Fig. 17).

Finally, using the same simulated dataset as before, we computed the correlation between each cell's expression profile and the bulk RNA-seq profiles for the 9 immune cell populations they were generated from. If the correlation between a cell and its corresponding bulk RNA-seq sample is larger than the others, we say it is correctly classified. We would expect that after imputation, the correlation between a cell and its corresponding bulk RNA-seq sample would be clearly larger than its correlation to the other samples, and hence the classification rate would improve. As shown in Supplementary Fig. 10A, most cells are classified correctly in the original data. ALRA successfully increases the correlation of each cell with its corresponding bulk RNA-seq sample more than with the other cell-type bulk profiles. DCA, MAGIC, scImpute, and SAVER, in contrast, do not improve the correlations; in fact, the misclassification error is higher after matrix completion using these methods.

We also noticed that ALRA appears to perform best on datasets that include a diversity of cell types. In the presence of diverse cells, the leading singular values, corresponding to biological variation, are large relative to the technical noise. Consequently, the corresponding singular vectors can be more accurately estimated, resulting in better imputation results. When we applied ALRA to the individual purified PBMC populations from above (Fig. 2), it did not preserve the biological zeros as effectively as when we applied ALRA to the merged matrix. In the mouse intestine cells, we similarly found that when ALRA is applied to a biological replicate that had the epithelial cells removed, the two populations of fibroblasts did not have as clearly distinguishable Pdgfra expression (Supplementary Fig. 18). For this reason, we recommend using ALRA on the full dataset, as opposed to purified subsets.

**ALRA is highly scalable**. ALRA only requires a single rank-$k$ SVD of the observed matrix, which is very fast to compute. Furthermore, it can scale to huge matrices with modern algorithms and software. In Fig. 4A we compare the runtime of ALRA, MAGIC, SAVER, DCA, and scImpute on subsets of mouse visual cortex cells from Hrvatin et al. The resulting matrices ranged from 1000 to 50,000 cells, each with 19,155 genes, and all methods were restricted to using only a single core. ALRA and MAGIC are significantly faster than the other methods, taking ~40 min on 50,000 cells. DCA requires about 4 h to run on the same number of cells, whereas scImpute and SAVER both take over 10 h for 10,000 cells and were not attempted on larger matrices. We emphasize that these are single-core times, several of the above methods can multithread, substantially improving performance on machines with multiple cores. ALRA also supports multithreading via the Intel Math Kernel Library

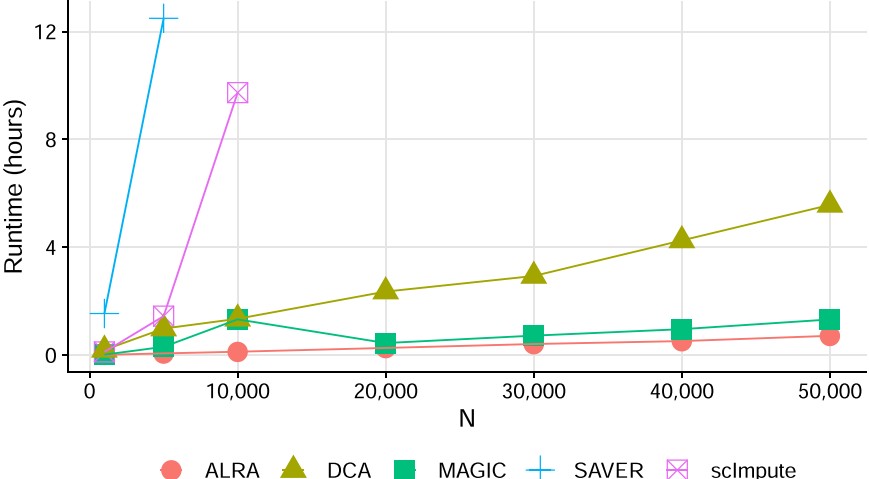

**Fig. 4 Runtime of ALRA versus other imputation methods using expression matrices of *N* cells and ~19,000 genes.** All experiments performed on a single core.

(MKL) by passing the 'use.mkl' flag to the ALRA function, but we do not compare multithreaded performance here. ALRA's scalability is especially crucial for the imputation of modern scRNA-seq datasets, which can exceed 1 million cells (e.g., ref. [29]).

In this paper, we present ALRA, an efficient imputation method for scRNA-seq data. ALRA's ability to preserve the vast majority of biological zeros rest on a solid theoretical foundation and its overall performance has been validated across simulated and real scRNA-seq datasets. ALRA improves the separation of cell types in both t-SNE and the original high dimensional space, imputes values consistent with external measurements, and computationally scales better than other methods. We also note that ALRA has only one parameter, the approximate rank *k* of the matrix, which is selected automatically based on statistics of the spacings between consecutive singular values.

## Methods

**ALRA procedure**. Given a measured expression matrix, as outlined in Algorithm 1, ALRA consists of the following steps. First, as commonly done with scRNA-seq data, the first step is standard pre-processing of the expression matrix by library normalization and log-transformation. Specifically, we normalize each column to $\alpha = 10,000$ transcripts per cell, add 1 to every entry (i.e., a "pseudocount"), and take the logarithm of each entry to obtain the normalized expression matrix. It has been noted that different values of $\alpha$ can affect downstream results[30], but we show that ALRA's preservation of biological zeros is not dependent on this choice (Supplementary Tables 6 to 9, Supplementary Fig. 19). As discussed in the Appendix, a variance stabilizing transform is necessary for good downstream results, and we chose the log-transform due to its widespread use. Other kinds of normalization can also likely be used.

After normalization, the next three steps are as follows: (i) Estimate the rank *k* (ii) compute the rank-*k* SVD using randomized SVD and (iii) threshold each gene of the rank-*k* SVD by the absolute value of that gene's $p = 0.001$ quantile. Finally, we rescale the resulting values such that the mean and standard deviation of the non-zero values of each gene in the resulting matrix match those of the normalized matrix (prior to low-rank approximation). A small number of non-zero values in the normalized matrix may be smaller than the threshold and may thus get set to zero by the thresholding operation in step (iii) of ALRA. For improved consistency with the observed matrix, we restore these entries to their original normalized values. We note that this process not only imputes technical zeros, but also denoises the originally non-zero values. An intuitive measure of how much the non-zero values were adjusted is the Pearson's correlation between these values before and after ALRA (e.g., for the purified PBMCs and Hrvatin et al. datasets, the correlations were 0.56 and 0.23, respectively).

Implementing ALRA using standard SVD procedures in common software may be rather slow and not scale to large matrices. To allow ALRA to scale to matrices with hundreds of thousands to millions of cells, we apply recently developed methods for randomized SVD[31]. Randomized SVD can be used to compute the leading singular values and singular vectors of large matrices with high accuracy yet at a fraction of the computing time and memory requirements. By default, ALRA uses the single-threaded implementation from the R package rsvd[32]. For Linux and OS X users, we also developed a multithreaded implementation based on Intel

Math Kernel Library (MKL) that can be installed with one command and enabled by the 'use.mkl' argument of the alra function. When imputing data using randomized SVD, we set the parameter for the number of additional power iterations to 10, for increased accuracy.

**Algorithm 1**. Adaptively thresholded Low-Rank Approximation

**Input:** Raw expression matrix $X$, number of rows (genes) $m$, number of columns (cells) $n$
**Output:** Imputed expression matrix $Y$

**Step 0.** (Pre-processing) Normalization
**for** $i \leftarrow 1$ **to** $n$ **do**
$\quad$ Normalize each column to $10,000$ transcripts: $X_{:,i} \leftarrow 10,000\,(X_{:,i}) \big/ \sum_{j=1}^{m} X_{j,i}$
$\quad$ Add a pseudo-count and take the logarithm: $X_{:,i} \leftarrow \log(X_{:,i} + 1)$
**end**

**Step 1.** Estimate $k$.
Compute the rank-100 SVD $\sum_{i=1}^{100} u_i v_i^T \sigma_i$ of $X$.
Define the spacings $s_i = \sigma_i - \sigma_{i+1}$.
Compute the mean $\mu$ and standard deviation $\sigma$ of $s_{79}, ..., s_{99}$.
Let $k = \max\{\, i \mid s_i > \mu + 6\sigma \,\}$.

**Step 2.** Imputation by low-rank approximation.
Compute the rank-$k$ randomized SVD $Y = \sum_{i=1}^{k} u_i' v_i'^T \sigma_i'$ of $X$.

**Step 3.** Restoration of biological zeros.
**for** $i \leftarrow 1$ **to** $m$ **do**
$\quad$ Let $c_i$ denote the $p = 0.001$ quantile of $Y_{i,:}$
$\quad$ **for** $j \leftarrow 1$ **to** $n$ **do**
$\quad\quad$ **if** $Y_{i,j} < c_i$ **then**
$\quad\quad\quad$ $Y_{i,j} \leftarrow 0$
$\quad\quad$ **end**
$\quad$ **end**
**end**

**Step 4.** Shifting and rescaling.
**for** $i \leftarrow 1$ **to** $m$ **do**
$\quad$ Let $\mu_i, \sigma_i$ denote the mean and standard deviation of the non-zero entries of $X_{i,:}$
$\quad$ Let $\mu_i', \sigma_i'$ denote the mean and standard deviation of the non-zero entries of $Y_{i,:}$
$\quad$ Shift and rescale the imputed values

$$Y_{i,:} = (Y_{i,:} - \mu_i') \cdot \frac{\sigma_i}{\sigma_i'} + \mu_i$$

**end**

**Rank estimation**. As described above, the only parameter in ALRA is the rank *k* of low-rank approximation. Estimating the rank of a matrix, given a noisy and/or corrupted version of it, is a classical and well-studied problem, see for example refs. [33–36]. Most works on rank estimation, however, assume that all entries of the matrix are observed, often with each entry corrupted by additive noise. In ref. [37], in an econometric context, an estimator was based on the spacing of eigenvalues. Specifically, the estimated rank is the largest value of *k* for such that $\lambda_k - \lambda_{k-1} > \delta$ for an appropriate threshold $\delta$, computed in a data-driven manner.

Motivated by this work, here we also propose a similar approach. Noise typically manifests as a long tail of singular values whose singular vectors point to random directions in space with no scientific significance. The goal is to choose *k* such that the singular values $\sigma_1, \ldots, \sigma_k$ correspond to biological variability of interest or signal, whereas the remaining values are noise. We assume an upper bound $k < 100$, and thus first use rsvd to compute the top 100 singular values. For this computation to be fast, we apply rsvd with the default setting of only $q = 2$

additional power iterations. Next, we compute the spacings between consecutive singular values, denoted as $s_i = \sigma_i - \sigma_{i-1}$ for $i = 2, \ldots, 100$. As can be seen in the plots of $s_i$ for two different datasets (Fig. 5), for large values of $i$, the gap between consecutive eigenvalues becomes very small, whereas the first few gaps are significantly larger. This is consistent with studies in random matrix theory considering spiked matrices containing signal+noise. Our aim is thus to choose the largest $k$ for which $s_k$ is significantly different from typical spacings between noise eigenvalues. To this end, we compute the mean $\mu$ and standard deviation $\sigma$ of the "noise" spacings $s_{80}, \ldots, s_{100}$. Next, our estimated value for the rank $k$ is the largest index for which the gap $s_k > \mu + 6\sigma$. A detailed theoretical justification for this procedure is beyond the scope of this manuscript. Yet, empirically, this method appears to be quite effective. Finally and importantly, we note that ALRA is not sensitive to the specific choice of a rank $k$: the number of biological zeros preserved does not significantly change with $k$ (Supplementary Table 4). Similarly, ALRA's improved separation of cell types is also robust to the exact value of the rank (Supplementary Fig. 13).

**Excess zeros**. After low-rank approximation, ALRA restores biological zeros by thresholding each gene based on the distribution of negative values. This allows the user to be confident that any imputed values do not correspond to biological zeros. However, values corresponding to technical zeros will inevitably be below the threshold and hence kept as zeros. This is why we consider ALRA to be a conservative method for imputation: zeros are only imputed if we have strong evidence that they are not biological zeros.

However, we can use the symmetry assumption to determine how many technical zeros are still missing after imputation. Given that the biological zeros are symmetrically distributed around zero, asymmetry in this distribution is due to technical zeros. This allows us to determine the number of excess zeros for a gene $i$ with threshold $\tau_i$ as

$$\psi_i = |\{j : 0 < y_{ij} < \tau_i \text{ and } x_{ij} = 0\}| - |\{j : y_{ij} < 0 \text{ and } x_{ij} = 0\}|,$$

where $x_{ij}$ and $y_{ij}$ denote the expression values before and after low-rank approximation, respectively, for the $i$th gene and $j$th cell. That is, the number of positive values set to zero in excess of the number of negative values set to zero. If there are no technical zeros below the threshold, then the number of positive values below the threshold (first term) will be the same as the number of negative values (second term), and hence $\psi_i$ will be close to zero. Conversely, if there are more positive values below the threshold than negative values, this suggests that a large number of technical zeros are also below the threshold will be set back to zero, as they cannot be distinguished from the biological zeros. To demonstrate this measure, we plot six genes with low $\psi$ and six genes with high $\psi$ (Supplementary Figs. 20 and 21). The ALRA package includes a function that can be used to calculate the number of excess zeros for genes of interest.

**Assumptions, theory, and relation to low-rank matrix completion**. The key assumption underlying ALRA is that the true scRNA-seq data matrix is low-rank, with the expression values lying in a linear subspace of dimension significantly lower than the number of cells or genes. It has long been appreciated[38–41] that genes do not act independently, but rather in concert, forming groups of highly correlated genes often referred to as gene modules. Therefore, the true expression matrix is often modelled as a low-rank matrix. Due to this assumed low-rank structure, indeed nearly every scRNA-seq pipeline first reduces dimensionality to the top few principal components (e.g., ref. [42]).

At first sight, imputing an scRNA-seq data matrix containing many zeros and assumed to be of low rank, may look like a particular instance of a low-rank matrix completion (LRMC) problem. This may seem a promising venue as over the past 10 years, a rich theory and several methods have been developed to complete missing values in low-rank matrices (e.g., refs. [43–45]). However, as we now explain, imputing scRNA-seq data are a different problem. Specifically, in the low-rank matrix completion literature, the assumption is that the set of indices of the observed entries are perfectly known, and have been sampled either uniformly at random, or as in ref. [46], with probability proportional to the leverage scores of the matrix. Under this sampling model, and some additional delocalization and incoherence assumptions, LRMC methods can exactly recover the true low-rank matrix with high probability. In scRNA-seq, in contrast, the precise indices where the values of the matrix were sampled are not perfectly known, since some of the observed zeros are true zeros of the underlying data matrix. Namely, the missing entries are some unknown subset of the observed zeros. Direct application of an LRMC method to scRNA-seq data (e.g., as in refs. [47,48]) while treating all the zeros as missing implies a highly biased sampling, which consequently may lead to incorrect imputation of the matrix. Moreover, the theoretical guarantees found in the LRMC literature do not apply to this biased sampling scheme.

To perform accurate imputation for scRNA-seq data, our method thus relies on a different yet simple observation: In datasets where the biological zeros were known, the non-zero values recovered by a low-rank SVD reconstruction of the observed matrix were symmetrically distributed around zero. In the supplement, we present the rigorous theoretical foundation for this empirical observation, relying on a perturbation analysis together with results from random matrix theory. Specifically, we prove that under certain assumptions, the distribution of these entries is indeed Gaussian with mean zero. Therefore, for each gene, any

element larger in magnitude than the $p = 0.001$ quantile is unlikely to be a biological zero. Conversely, values smaller in magnitude than this threshold may be biological zeros, so we set them to zero. The result is a completed matrix where the biological zeros are preserved, with high probability, and where every imputed value is unlikely to be a biological zero. We note that ALRA is not as sensitive to the magnitude of $p$ (Supplementary Fig. 14).

We remark that in our theoretical analysis, we assumed that the singular values of the low-rank matrix are sufficiently large to be detected by our rank estimation procedure, even in the presence of a significant number of technical zeros. In the theory of high dimensional statistics concerning signal+noise matrices, there is a well-studied phase transition, whereby weak signals cannot be reliably detected by SVD (or PCA), and then the sample eigenvectors bear no relation to the true population ones[49,50]. In our context, in scRNA-seq experiments involving several types of cells, we expect the top eigenvectors to capture the differences between different cell types, and thus leading to large singular values, relative to the noise. In contrast, in scRNA-seq data containing only a single type of cell, the often smaller cell-to-cell heterogeneity between cells of the same type may lead to weak signals that due to a large number of technical zeros, cannot be discovered by SVD. In such a case, ALRA may consequently fail to correctly detect and preserve the biological zeros.

**Purified PBMC analysis**. Filtered expression matrices were obtained for FACS purified B cells, CD14 monocytes, CD34+ cells, CD4 helper T cells, regulatory T cells, naive T cells, memory T cells, CD56 natural killer cells, cytotoxic T cells, and naive cytotoxic T cells from the 10X Genomics website. The matrices were merged, resulting in an expression matrix with 94,655 cells and 32,738 genes. The matrix was filtered so that cells expressing more than 400 genes and genes expressed in more than 100 cells were retained, resulting in a matrix with 83,992 cells and 12,776 genes. We applied DCA, SAVER, and scImpute to this count matrix (as these methods take the raw data and perform their own normalization), and for SAVER we sampled from the estimated posterior distribution using the function 'sample.saver'. We ran ALRA and MAGIC on the library and log normalized the expression matrix. The outputs for DCA and scImpute were log and library normalized. The output of SAVER was log normalized but not library normalized, as it was not run on all genes (see below). The rank for ALRA was estimated to be $k = 26$ using the procedure described above. For MAGIC, we set the negative values to be zero.

Due to computational constraints, we cannot run SAVER and scImpute on all the genes. Using SAVER's 'pred.genes' argument, we only imputed the genes identified as containing biological zeros, making it computationally feasible. scImpute does not have such an argument, so to compute the proportion of zeros preserved and completed, we ran scImpute on a random set of 10,000 cells. To compute the proportion of total zeros completed (TC) in Fig. 2C for SAVER, we also ran SAVER again on all genes using the same random subset of 10,000 cells as before.

**Simulated scRNA-seq data**. To evaluate ALRA's performance, we simulated scRNA-seq data based on deep-sequenced bulk RNA-seq data of $S$ samples. The simulation is based on a multinomial model, as in refs. [17,18]. Enumerating the bulk RNA-seq expression profiles as $i = 1, \ldots, S$ and genes as $j = 1, \ldots, m$, let $n_j^i$ denote the read counts of gene $j$ in the $i$th sample. Normalizing, we parameterize the multinomial distribution with an $m$-dimensional probability vector $p^i$, where

$$p^i(j) = \frac{n_j^i}{\sum_{k=1}^m n_k^i}.$$

To generate the gene expression profile for the $\ell$th cell, we randomly choose one of the cell types with equal probability and denote the chosen cell type by $c_\ell \in \{1, \ldots, S\}$. The associated probability vector is then $p^{c_\ell}$. In order to obtain a realistic distribution of read counts, we randomly sample a cell from the purified PBMC dataset, and let $N_\ell$ be the read count in that cell. Finally, we sample an $m$-dimensional vector of counts from the multinomial distribution parameterized by $N_\ell$ and $p^{c_\ell}$. The process is then repeated for as many cells as specified by the simulation.

This simulation data enables us to distinguish true biological zeros and technical zeros based on whether or not the gene is actually expressed in the corresponding bulk RNA-seq sample. For Fig. 2D, we used 5 spleen T-cell types (Treg, CD4 conventional T cells (TConv), CD8 T, gamma delta T cells (gdT), and Natural Killer T cells). To determine the imputation methods' performance as a function of increasing read depth, we varied read counts as $\alpha N_\ell$, where $\alpha = 0.25, 0.5, 0.75, 1.0$.

For the correlation study in Supplementary Fig. 10 we used a similar multinomial-based sampling approach, but we used all nine cell types (CD4 conventional T cells (TConv), CD8 T, gamma delta T cells (gdT), neutrophil, NK, NKT, and Treg cells).

**Evaluation of zero preservation with bulk RNA-seq data**. To comprehensively evaluate the ability of various imputation methods to preserve biological zeros, we used bulk RNA-seq data to define biological zeros. In ref. [14], several different types (B cells, monocytes, T cells, and NK cells) of peripheral blood mononuclear cells (PBMCs) were analyzed with bulk RNA-seq, with two biological replicates for each.

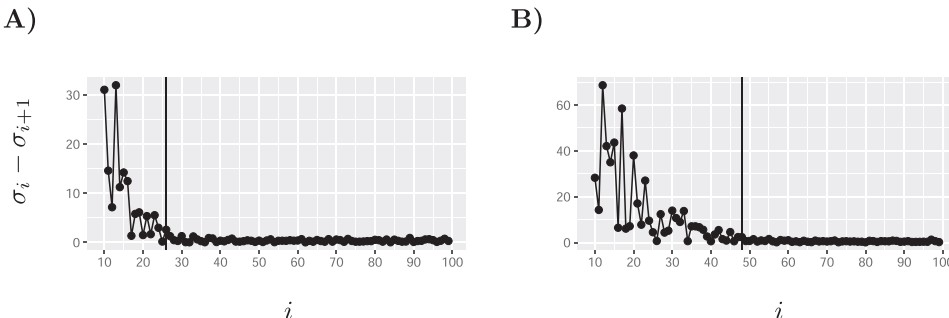

**Fig. 5 Differences between consecutive singular values. A** the rank was set to $k = 26$ in the purified PBMCs experiment. **B** Rank was set to $k = 48$ for the mouse visual cortex cells experiment.

For each cell type, genes with zero counts in both replicates were defined as "biological zero genes" for this specific cell type. In the scRNA-seq data of the corresponding cell type, we calculate the zero preservation rate as a number of zero entries in all biological zero genes after imputation, divided by a number of zero entries in all biological zero genes before imputation.

**CITE-seq data analysis**. Cellular Indexing of Transcriptomes and Epitopes by sequencing (CITE-seq)[15] is a method for simultaneous epitope and transcriptome measurement in single cells. We obtained the PBMC dataset from GEO (GSE100866 [https://www.ncbi.nlm.nih.gov/geo/query/acc.cgi?acc=GSE100866]) and proceeded to identify cell types based on their surface protein markers. After filtering cells for which more than 90% of the UMIs came from mouse genes and removing genes that had expression in less than 10 cells, there were 7667 cells and 13,458 genes. Surface protein markers are measured by counting antibody-derived tags (ADT) on each cell, and gene expression is measured in the same cells using scRNA-seq. We used the ADT counts to determine cell type by "gating" the cells by their marker epitopes. We used markers CD3, CD19, and CD14 to define T cells, B cells, and monocytes, respectively. Specifically, we applied $k$-means with $k = 2$ to transformed ADT counts for each marker individually in order to classify cells as expressing a marker epitope or not. Then, we classify cells into types as follows: CD3+CD19−CD14− T cells, CD3−CD19+CD14− B cells, and CD3−CD19−CD14+ monocytes. We ran ALRA and the other imputation methods on the scRNA-seq data; the rank for ALRA was estimated to be $k = 22$ using the procedure described above.

We obtained the cord blood mononuclear cell (CBMC) dataset from the same paper and imputed the data using ALRA and the other imputation methods. After filtering cells for which more than 90% of the UMIs came from mouse genes, there were 8005 cells and 20,400 genes. For ALRA, the rank was estimated to be $k = 25$. We used CLR-transformed epitope ADT counts and calculated the correlation between each protein marker and its corresponding gene, before and after imputation. We also calculated the correlation between each protein marker and 1000 random genes and compared the correlation of each protein marker and its corresponding gene to this "null distribution". We also computed the Euclidean distances between each pair of cells in the ADT space and then in the gene expression space, and compared the correlation of these distances before and after imputation.

**Processing of mouse dermal cells**. Raw count scRNA-seq data of E14.5 Wild Type (WT: two replicates) and Loss of Function (LOF: one replicate) was obtained from[4]. For each sample, we used Seurat to perform data normalization, scaling, variable gene selection, PCA, clustering, and t-SNE embedding calculation. Cells with expression in less than 1000 genes were filtered out. The *Col1a1* gene was used to select dermal clusters, and only cells in clusters with high expression (>85% expressing cells) of *Col1a1* were retained for further analysis. The resulting matrix had 11,121 dermal cells and 28,000 genes. After the selection of dermal cells, we first merged the two biological replicates for the WT samples, then used canonical correlation analysis (CCA) to merge data from WT and LOF and removed batch effect. The top 15 CCA components were used for t-SNE calculation. The expression matrices were normalized as in the Purified PBMC analysis detailed above. The rank for ALRA was estimated to be $k = 49$ using the procedure described above.

**Identification of PBMCs using known markers**. The 68k peripheral blood mononuclear cell dataset was downloaded from 10X Genomics website[9]. Cells with expression in less than 400 genes were filtered out, and genes with expression in less than 6 cells were filtered out. The resulting dataset had 58,674 cells and 16,137 genes. The rank for ALRA was estimated to be $k = 19$ using the procedure described above, and ALRA was run on the dataset. Positive and negative markers were determined based on ref. [19] and supplemented with expert knowledge (Supplementary Table 5). If a cell expressed all positive markers and no negative

markers after imputation, then it was identified as belonging to that corresponding class.

**Processing of data on human bronchial epithelial cells**. Count matrices after filtering were downloaded from Lukassen et al. As described in the original paper, Seurat (Version 3)[51] was used to perform library and log normalization, scaling, integration, variable gene selection, PCA, and UMAP embedding calculation with default parameters. Cell-type information was also obtained from the original work. ALRA was applied on the merged and normalized data matrix.

**Improved separation of cell populations**. The scRNA-seq expression matrix of mouse visual cortex cells from ref. [25] was obtained and filtered as in Huang et al., so as to compare with SAVER's result. Specifically, genes with a mean expression less than 0.00003 and non-zero expression in less than 4 cells were excluded, resulting in an expression matrix with 19,155 genes and 65,539 cells. ALRA was run with $k = 48$ (as estimated by the procedure described above) on the subset of 48,244 cells that were classified into cell types in the original study. Variable genes were identified for both the observed and ALRA-completed data using Seurat[52]. The fast t-SNE implementation of ref. [53] (FIt-SNE) was then used to compute the embedding of the top 48 principal components of the observed and ALRA-completed data.

Next, the scRNA-seq data[26] of neocortical cells were obtained and analyzed. We filtered cells labeled with 'Low Quality', 'Doublet', 'Batch Grouping', or 'High Intronic', based on the metadata provided by the authors, and there were 21,874 cells left. ALRA, DCA, and MAGIC were run on the data. The rank $k$ for ALRA was estimated to be $k = 53$. On each method, we then used Seurat to scale the data, identify variable genes, compute PCA, and computed t-SNE using the top 53 PCs.

We also used the downsampled scRNA-seq datasets[21–24] from ref. [7]. Cell identity was based on the metadata provided by the original authors. Baron et al. had 1076 cells, Chen et al. had 7712 cells, La Manno et al. had 947 cells, and Zeisel had 1799 cells. When running ALRA on the four datasets, the $k$s for ALRA were automatically chosen to be 12, 21, 15, and 16 respectively. After imputation with each method, the number of PCs to retain for each was chosen by the jackstraw method as implemented in Seurat. PCs with an assigned $p$-value of $1 \times 10^{-5}$ or smaller were retained.

We then sought to quantify the separability of cell types after imputation. For each dataset, we trained a random forest to classify cell type based on expression values (using the raw expression matrices obtained from ref. [7]). The classification error is then a surrogate for separability, as better separated cell types are easier to classify. We used the R package ranger[54] with default settings and compared the out-of-bag error before and after imputation.

Similarly to ref. [25], for the four matrices in the downsampling experiment, we computed the Spearman's correlation between the genes/cells in the imputed matrix and genes/cells in the original (i.e., not downsampled) matrix. Instead of Pearson's rho, we used Spearman's correlation as it is more robust to the presence of strong outliers that we observed. We then computed PCA for each of the datasets, and retained the leading PCs, which were chosen using Seurat's jackstraw command on the original matrix. Finally, we computed the subspace angle between the PCs of each original dataset and the PCs of the downsampled matrix before and after imputation. The subspace angle was calculated using the subspace function in the R library pracma[55].

**FISH experiments**. We obtained the Drop-seq and RNA FISH datasets from[27]. The drop-seq data consists of 8640 cells and 32,287 genes. The RNA FISH consists of 26 genes across 7000–88,000 cells (depending on the gene). To directly compare with analysis published by the authors of SAVER, we followed their same filtering procedure, removing cells with library size greater than 20,000 or less than 500, and also removing genes with mean expression less than 0.1. The resulting dataset contains 8498 cells and 12,241 genes, with 16 genes shared between Drop-seq and FISH. As in Huang et al., cells in the bottom and top tenth percentiles of the housekeeping gene *GAPDH* were filtered out, and all genes were then normalized

by *GAPDH* expression. We then computed the Gini coefficient of each of the other 15 genes (excluding *GAPDH*) in RNA FISH, original Drop-seq, and Drop-seq after imputation, using this filtered subset (Supplementary Fig. 12). The rank selected by ALRA was $k = 14$.

**Runtime comparison**. We sampled random subsets of sizes 5000, 10,000, 20,000, 30,000, 40,000, and 50,000 cells from Hrvatin et al. and compared the runtime of ALRA and other imputation methods on each subset. All experiments were performed on a single core, with default parameters. Notably, SAVER's option of 'do.fast' was set to true.

**Intestinal fibroblast experiments**. Pdgfra[EGFP/+] mice[56] were obtained from the Jackson Laboratory and bred in the facilities of the Yale Animal Resources Center and maintained on a C57BL/6J genetic background. Mice were housed in standard cages, on a 12 h day/night cycle and were fed a standard rodent chow. All animal experimentation was performed in compliance with Yale Institutional Animal Care and Use Committee protocols.

*Isolation of mouse intestinal mesenchymal cells*. The intestine was dissected, flushed, opened longitudinally, and then cut into 1 cm pieces. The tissues were incubated in HBSS containing 1 mM EDTA, 1 mM DTT, 0.2% FBS, 4–5 times, 10 min each, at 37 °C, 200 rpm. Epithelial cells were released by vigorous shaking. After epithelial cell removal, the remaining stromal part of the intestine was incubated in DMEM 10% FBS containing Collagenase XI (300 units/ml, Sigma, C7657), Dispase II (0.1 mg/ml, Sigma, D4693), and DNase II Type V (50 units/ml, Sigma, D8764) for 1 h, at 37 °C, 200 rpm. Cells released after vigorous shaking were passed through a 70 μm strainer and washed with 2% sorbitol. Such cell preparations were directly processed by Drop-seq or by flow cytometry.

*Flow cytometry in intestinal mesenchymal cells*. Flow cytometry analysis was performed at the Yale Flow Cytometry Facility. Freshly isolated intestinal mesenchymal cells from Pdgfra[EGFP/+] mice[56] were analyzed for direct EGFP fluorescence with a BD LSRII cytometer equipped with FACSDiva software. Cells isolated from the intestine of WT mice were used as a non-fluorescent control. Data analysis was performed with the FlowJo software. Single cells were selected first by a FSC-H vs FSC-A and then by a SSC-H vs SSC-A gate and then a FSC-A vs SSC-A gate was applied.

*Drop-seq in intestinal mesenchymal cells*. Mesenchymal cells were isolated from the intestine of WT C57BL/6J mice ($n = 2$). The cells were diluted to a concentration of 100 cells/μl and 1 ml aliquots were used as input to the Drop-seq protocol which was performed as described previously[42,57] with minor modifications. The beads were purchased from ChemGenes Corporation, Wilmington MA (catalog number Macosko201110) and the PDMS co-flow microfluidic droplet generation device was generated by Nanoshift, Emeryville CA. For both conditions, the 1 ml DropSeq collection was performed. Samples were processed for cDNA amplification within ~15 min of collection. Populations of 5000 beads (~150 cells) were separately amplified for 15 cycles of PCR (conditions identical to those previously described) and pairs of PCR products were co-purified by the addition of 0.6x AMPure XP beads (Agencourt). The cDNA from an estimated 1500 cells was prepared and tagmented by Nextera XT using 1000 pg of cDNA input, and the custom primers P5_TSO_Hybrid6[57] and Nextera XT primers, N701 and N702 (Illumina). Both libraries were sequenced on the Illumina NextSeq 500 using 2.0 pM in a volume of 1.3 ml HT1, and 2 ml of 0.3 μM Read1CustomSeqB6[57] for priming of read 1. Read 1 was 20 bp; read 2 (paired-end) was 60 bp. Single-cell RNA-seq data were processed as described by Macosko et al. to generate a digital expression matrix with transcript count data. This matrix was filtered retaining cells with more than 1000 transcripts and less than 10% mitochondria transcripts. The expression matrices were normalized as in the Purified PBMC analysis detailed above, except that SAVER was applied to all genes, and hence it was also library normalized (unlike in the purified PBMC analysis).

**Reporting summary**. Further information on research design is available in the Nature Research Reporting Summary linked to this article.

## Data availability

The mouse colonic mesenchyme Drop-seq data generated in this study have been deposited in the Gene Expression Omnibus database under accession code GSE185638 https://www.ncbi.nlm.nih.gov/geo/query/acc.cgi?acc=GSE185638. All other datasets are publicly available. Specifically, the purified PBMCs of Zheng et al. can be downloaded from 10x genomics website (https://www.10xgenomics.com/resources/datasets/). Bulk RNA-seq data from ImmGen (Heng et al.) can be downloaded from NCBI BioProject PRJNA281360. Other datasets can be downloaded from NCBI Gene Expression Omnibus: Hoek et al. (GSE64655), Stoeckius et al. (GSE100866), Gupta et al. (GSE122043), Hrvatin et al. (GSE102827), Tasic et al. (GSE115746), Torre et al. (GSE99330). The datasets from Chen et al. (GSE87544), Baron et al. (GSM2230757), La Manno et al. (GSE76381), Zeisel et al. (https://linnarssonlab.org/cortex) were used as preprocessed by Huang et al. and can be downloaded from https://github.com/mohuangx/SAVER-paper.

## Code availability

An R implementation of ALRA is freely available at https://github.com/KlugerLab/ALRA[58]. The codes for analyses this paper are available at https://github.com/KlugerLab/ALRA-paper[59].

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

## Acknowledgements

We would like to thank Stefan Steinerberger, Jeremy Hoskins, Jonathan Levinsohn, Ofir Lindenbaum, Dmitry, Kobak, Peggy Myung, and Holly Steach for many useful discussions. G.C.L. acknowledges support by the NIH NHGRI Grant #F30HG010102. Y.K. acknowledges support by NIH grant # R01GM131642, R01GM135928, UM1DA051410, P50CA121974, U01DA053628 and R01HG008383.

## Author contributions

G.C.L., J.Z., B.N., and Y.K. conceptualized the study; G.C.L. and J.Z. performed the computational experiments; M.R., P.B., and R.A.F. conducted the mouse sequencing experiment; M.R. and R.A.F. conducted the flow cytometry validation experiment; and G.C.L., J.Z., B.N., M.R., R.A.F., and Y.K. wrote the paper.

## Competing interests

R.A.F. is a consultant for GlaxoSmithKline and Zai Lab Ltd. P.B. is an equity holder in Celsius Therapeutics. The remaining authors declare no competing interests.
