## [Peer Review File · Nature Communications]

Title: Zero-preserving imputation of single-cell RNA-seq dataREVIEWER COMMENTS

Reviewer #1 (Remarks to the Author):

Review

The authors make use of PCA (SVD) to predict smoothed gene expression levels from scRNA-seq data. They apply a quantile cutoff on the predicted expression level to censor data as true zeros. This is an interesting flip on the much-discussed imputation strategy in scRNA-seq. Imputation methods typically aim to recover true hidden expression levels, while the aim here is to learn which genes are truly not expressed, with potential applications being negative marker identification or inferring suppressive regulation.

They use bulk RNA-seq as evidence that some genes are truly not expressed in some tissues. In particular, the authors use a good strategy to generate positive control data by sampling expression levels from ImmGen bulk RNA-seq data of sorted cell types. This way they know true and false zeros from sampling zeros.

Using known negative markers such as NCAM1 in B cells is also a clever strategy. However, it is worth keeping in mind that most evidence for such negative markers is based on fluorescence (FACS or qPCR) or electrophoresis. In these cases there is a positive background signal, and concluding a complete lack of a given molecule is difficult.

The use of the conditional knockout experiment data is a good way to handle this problem. However, in the text it is not clear which gene is knocked out and how the effect is reflected in the data.

It is a challenge to find proper control data for scRNA-seq data. Here CITE-seq counts and smFISH counts are used. However, these suffer from similar technical challenges as scRNA-seq (they are simply less discussed for cultural reasons). The protein counts in CITE-seq are more deeply sequenced than the RNA counts, but suffer from the same shortcomings as scRNA-seq with the addition of background from nonspecific antibody binding and inefficiencies (not to mention that that it's not clear that the two modalities should reflect each other on an individual cell level). While smFISH is estimated to be 80% efficient compared to the 2%-50% efficiency of scRNA-seq (depending on protocol and implementation), it also suffers from challenges with accurately segmenting cells and counting puncta.

The authors point out that some cells where genes that are typically annotated as unique markers for cell types have also been observed in other cell types as an argument for predicted expression in unexpected cells (NCAM1 in cytotoxic T cells). Another fundamental issue with this task however is the "soup-problem". In the vast majority of scRNA-seq methods cells are isolated at random with a cell suspension. This cell suspension will also contain a solution of mRNA from cells which have been damaged. So if cells are isolated in droplets (such as in 10x Chromium) these droplets will also contain mRNA from arbitrary cells, similar to a bulk RNA-seq sample. With this in mind, is there any hope that

true zeros will ever be expected between cell types in the sample? There are some proposed methods to filter the background 'soup' mRNA, but none of them are convincing.

When discussing expression levels there is a general lack of units. In particular, it does not appear units are comparable between the different methods tested.

The value of absolutism regarding 0-values being preserved is questionable. An absolute 0 chance of seeing a molecule in a cell from an experiment is an extremely strong prior. For example, even though DCA doesn't produce 0-values, since it uses a statistical model for counts, the output (expected counts through the ZINB parameters) can be directly converted to a probability of seeing a count > 0 (given some sequencing depth).

Outside of directly investigating if a gene is lacking the preservation of zeros seems unlikely to be useful. There are no downstream methods which can properly handle these zeros, which is why so many methods attempt to impute zeros to real values. Investigation on how reconstructed expression levels affect the various tSNE analysis adds little information of interest: it is highly likely even better performance would be achieved if not censoring some values to 0, or even if the tSNEs were applied to the principal components from the SVD/PCA rather than first reconstructing the matrix.

Reviewer #2 (Remarks to the Author):

In this paper, the authors developed a method called ALRA that imputes single-cell RNA-seq data using singular-value decomposition (SVD) plus thresholding. They demonstrated the performance of ALRA using three real datasets and three simulation studies (synthetic data), and they downsampled the original data to create the data that needed imputation and evaluated the imputation results by the original data.

I believe this method would be useful for large-scale scRNA-seq datasets given its good computational efficiency, if the validation results were convincing. The authors showed that ALRA has improved accuracy measured by the Gini index, which benchmarks imputed scRNA-seq data against FISH data, over that of the existing imputation methods. However, I have the following major concerns regarding the validity of the results. Most importantly, the authors need to provide more detailed figure captions to make the figures understandable.

1. In the section "ALRA preserves biological zeros in simulated and real datasets," the authors defined biological zeros based on "genes with solid biological knowledge not expressed in a cell population." However, the authors did not provide literature or data evidence to justify their definitions, that is, what genes are not expressed in what cell types by what evidence.

2. In the analysis of "The importance of preserving biological zeros: an example", SAVER and scImpute

were no longer compared. The authors should include these two methods to be consistent with other results.

3. It is unclear why ALRA favors diverse cell types. This important phenomenon needs to be justified.

4. The choice of the $p = 0.001$ quantile seems arbitrary and unjustified. The choice of this threshold is crucial to the performance of ALRA, so the authors need to provide a practical guide for users to make the choice a data-driven way.

5. Many figures have vague captions and labels. For example, Figure S8 is hard to understand because the meaning of the y-axes is unclear. The message in Fig. S3 is also unclear.

6. In the section “ALRA improves separation of cell types,” the authors claimed that “To quantitatively confirm that ALRA increases separation, we trained supervised random forests to classify the cells into cell types based on expression values”. However, the SAVER paper argued the other way: the cell-types labeled in Zeisel et al. were based on cell clusters found from the raw scRNA-seq data and not trustworthy, so imputation was needed. In fact, the SAVER paper used the cell clusters found from their imputed Zeisel et al. data instead of the cell types for evaluation. Given the argument in the SAVER paper, the use of cell types to train a supervised classification algorithm is unconvincing. The authors need to justify it.

RESPONSE TO THE REVIEWERS' COMMENTS

REVIEWER 1

The authors make use of PCA (SVD) to predict smoothed gene expression levels from scRNA-seq data. They apply a quantile cutoff on the predicted expression level to censor data as true zeros. This is an interesting flip on the much-discussed imputation strategy in scRNA-seq. Imputation methods typically aim to recover true hidden expression levels, while the aim here is to learn which genes are truly not expressed, with potential applications being negative marker identification or inferring suppressive regulation.

As the reviewer has pointed out, a major advantage of ALRA over existing methods is the preservation of biological zeros. In fact, this was our inspiration for developing ALRA: practitioners reported that they were uncomfortable using existing imputation methods that incorrectly imputed genes as being expressed in cells where they were known to not be expressed.

However, we emphasize that ALRA's utility is not restricted to inference of true zeros, such as in identifying negative markers or inferring suppressive regulation. The thresholding operation at the heart of ALRA is a well-established strategy for denoising (e.g. landmark papers by David Donoho and Iain Johnstone on soft thresholding). Of note, even if there were no biological zeros in the true, underlying expression matrix (i.e. all genes have a small probability of being expressed), the thresholding strategy would still improve downstream analyses. This is because thresholding is minimax rate optimal for approximately sparse signals measured with noise. Further, we establish theoretical guarantees for ALRA's performance, in stark contrast to the heuristic nature of existing methods.

The advantages of this denoising operation are reflected in various downstream analyses. For example, we showed that ALRA improves separation of cell types in six different real-life scRNA-seq datasets, recovers true cell-to-cell distances as validated using CITE-seq data, and improves correlation of simulated expression profiles to known references. This is to say that ALRA is not only superior to existing methods in terms of preserving zeros, but also in recovering true hidden expression levels.

They use bulk RNA-seq as evidence that some genes are truly not expressed in some tissues. In particular, the authors use a good strategy to generate positive control data by sampling expression levels from ImmGen bulk RNA-seq data of sorted cell types. This way they know true and false zeros from sampling zeros. Using known negative markers such as NCAM1 in B cells is also a clever strategy. However, it is worth keeping in mind that most evidence for such negative markers is based on fluorescence (FACS or qPCR) or electrophoresis. In these cases there is a positive background signal, and concluding a complete lack of a given molecule is difficult.

Every gene may have a non-zero probability—however small—of being expressed in any cell. However, there is substantial literature, particularly relating to flow cytometry, where cell types are defined by the presence or absence of certain markers. In this literature, genes expressed at negligibly low levels are considered to be “not expressed” in a certain cell type. Put differently, genes that are expressed at levels below the noise level (the positive background signal mentioned by the reviewer) are considered to be not expressed. Unlike FACS and qPCR that require manual identification of the noise level, ALRA denoises by automatically thresholding based on an adaptive bound for the noise in each cell. After the thresholding operation, the remaining non-zeros values represent true expression with high probability.

The use of the conditional knockout experiment data is a good way to handle this problem. However, in the text it is not clear which gene is knocked out and how the effect is reflected in the data.

We replicated an analysis by Gupta et al. of mouse embryonic skin cells to demonstrate the importance of preserving biological zeros. A major finding of this study was that Wnt signalling is necessary for differentiation of dermal condensate (DC) cells. The authors conditionally knocked down the Wnt pathway and then showed that only dermal cells retaining Wnt activity were able to develop into DC cells. A key step in their analysis was quantifying the proportions of cells which express DC cell markers and Wnt signalling markers. Using other imputation methods that do not preserve biological zeros, the practitioner would have to manually threshold the expression values to determine which cells express these markers. Manual thresholding can be arbitrary and lead to erroneous results. In contrast, ALRA does not require this thresholding step, as cells not expressing the gene will not be falsely imputed to non-zero values.

In the revised version of the paper, we give more details about the knockout experiment. Specifically, the Wnt pathway was made inoperative by conditionally knocking out exons 2-6 of beta-catenin (Ctnnb1). While this renders beta-catenin ineffective, it is still detected by scRNA-seq as the 3'-end is not affected (i.e. the exon adjacent to the 3'-end is not knocked out). Therefore, the downstream genes Lef1 and Axin2 are used as surrogates for Wnt activity instead of directly measuring beta-catenin expression.

It is a challenge to find proper control data for scRNA-seq data. Here CITE-seq counts and smFISH counts are used. However, these suffer from similar technical challenges as scRNA-seq (they are simply less discussed for cultural reasons). The protein counts in CITE-seq are more deeply sequenced than the RNA counts, but suffer from the same shortcomings as scRNA-seq with the addition of background from nonspecific antibody binding and inefficiencies (not to mention that that it's not clear that the two modalities should reflect each other on an individual cell level). While smFISH is estimated to be 80% efficient compared to the 2%-50% efficiency of scRNA-seq (depending on protocol and implementation), it also suffers from challenges with accurately segmenting cells and counting puncta.

We agree with the reviewer that CITE-Seq and smFISH each have their own technical challenges and can certainly not be considered a "gold standard" for scRNA-seq. However, we believe that improved consistency of the scRNA-seq expression data with these other methods indicates that the imputed values are more reflective of biology.

We opted to use CITE-Seq and FISH to validate our results, as done by authors of other scRNA-seq imputation methods. Specifically, the FISH validation is exactly replicated from the SAVER paper (Huang et al. 2018, Nature Methods), and CITE-seq was also used by Eraslan et al. (2019, Nature Communications) to show that DCA increases protein and RNA co-expression. Given this existing precedent, we included these analyses to show that ALRA not only preserves biological zeros, but also outperforms other methods on metrics proposed by their original authors.

Finally, we emphasize that our goal was to use a variety of methods to validate ALRA including simulations, expert knowledge, subsampled scRNA-seq datasets, FISH, bulk RNA-seq, and CITE-seq datasets. While each validation method has its caveats, by considering these various methods, we can conclude that ALRA's consistently superior performance is unlikely to be artefactual as their uncertainties are complementary.

The authors point out that some cells where genes that are typically annotated as unique markers for cell types have also been observed in other cell types as an argument for predicted expression in unexpected cells (NCAM1 in cytotoxic T cells). Another fundamental issue with this task however is the ‘‘soup-problem’’. In the vast majority of scRNA-seq methods cells are isolated at random with a cell suspension. This cell suspension will also contain a solution of mRNA from cells which have been damaged. So if cells are isolated in droplets (such as in 10x Chromium) these droplets will also contain mRNA from arbitrary cells, similar to a bulk RNA-seq sample. With this in mind, is there any hope that true zeros will ever be expected between cell types in the sample? There are some proposed methods to filter the background ‘soup’ mRNA, but none of them are convincing.

In a typical scRNA-seq dataset, roughly 95% of entries are zero (and hence not affected by the soup problem), and roughly 5% are non-zero. Within those 5%, some false non-zero values are due to contamination by ambient RNA (i.e. the soup problem); correcting this measurement error is an interesting area of investigation.

We did a simple experiment to quantify the effect of the soup problem in the purified PBMC datasets where we define true biological zeros using the corresponding bulk RNA-seq data. Only 0.1-0.3% of these biological zeros were non-zero in the original dataset, which is comparable to the purity of the cell population as measured by FACS. Overall, of the 21,808,779 known biological zeros across four datasets, at most 30,927 (0.142%) were affected by the soup problem. Given that the number of entries affected by the soup is so few, erroneous imputation based on these values is unlikely. Specifically, the effect on leading eigenvectors in the low-rank approximation would be negligible. For example, a small number of cytotoxic T-cells expressed CD4 in the original data (possibly due to the soup problem), and after imputation this number remained small (Figure 2B).

When discussing expression levels there is a general lack of units. In particular, it does not appear units are comparable between the different methods tested.

When using each imputation method, we attempted to follow the same preprocessing steps as the original authors. For DCA and scImpute, the input and outputs are read counts. In these cases, we normalize the outputs as usual: log and library normalization. SAVER also outputs read counts, but for large matrices where SAVER cannot be run on all genes, we only do log normalization. For MAGIC and ALRA, we use log and library normalization prior to imputation. As a result, all methods are log and library normalized. We make this clear in the revised version.

We also point out that our goal with showing the distributions of ALRA-imputed values in Figure 2A is to demonstrate that the negative values after low-rank approximation can be used to threshold the noise. In this setting, the units are not important; the relative size of the noise and true expression values is the main point.

The value of absolutism regarding 0-values being preserved is questionable. An absolute 0 chance of seeing a molecule in a cell from an experiment is an extremely strong prior. For example, even though DCA doesn't produce 0-values, since it uses a statistical model for counts, the output (expected counts through the ZINB parameters) can be directly converted to a probability of seeing a count > 0 (given some sequencing depth).

The reviewer makes an excellent point: perhaps a cell type that is reported to not express a given gene actually expresses it at very low levels (or a small subpopulation of cells express it). In the noiseless setting, assigning a low probability of expression to these values would be ideal. However, these small values are often unable to be estimated, as they fall below the noise level. ALRA identifies the size of entries that cannot be due to noise, and sets values below that level to zero.

In the manuscript, we demonstrate that DCA's reports unrealistically high values for these "biological zeros." For example, in Figure S3A we see that in CD8+ T-cells, the probability of CD4 expression is about 20% of CD8 expression! While it is possible that some small fraction of CD4 T cells express CD8 as well, this high level of expression is inconsistent with prior knowledge. The non-zero values which competing methods produce are not small, and hence their interpretation under a probabilistic model are not realistic.

We note that there are scenarios where absolutism regarding 0-values is well justified. One potential example may arise when analyzing pre- and post-treatment samples from cancer patients. Suppose a subset of cells in the tumor carry a particular mutation that can be probed by scRNA-seq technology. After treatment, it may be important to determine the residual tumor burden by estimating the fraction of cells that carry this mutation. By using an imputation method that preserves biological zeros, one can make the important clinical distinction between no expression (no disease) and minimal expression (minimal residual disease). We note that use of scRNA-seq (as opposed to DNA or bulk RNA sequencing) in this example may be more sensitive in the setting where tumor burden is low.

Outside of directly investigating if a gene is lacking the preservation of zeros seems unlikely to be useful. There are no downstream methods which can properly handle these zeros, which is why so many methods attempt to impute zeros to real values.

In the revised paper, we have added two examples of downstream analyses which are made possible by ALRA's strategy of assigning biological zeros by thresholding at the noise level. We show how ALRA can be used to *automatically gate* cells, taking advantage of a rich literature on negative and positive markers for known cell types. We used the markers published in The Human Blood Atlas (Uhlen et al. 2020) to identify known cell types in the peripheral blood monocytes (PBMC) scRNA-seq dataset (Zheng et al. 2018), such as identifying CD27+CD19+CD14-CD3G-cells to be Memory B cells. This is especially useful for identifying known populations of cells that are not well-separated in clustering or embeddings. For example, Naive B cells and Memory B cells appear to be a single cluster on the t-SNE, but our gating approach identifies these known subpopulations. By preserving biological zeros, ALRA does not require the user to make arbitrary thresholds defining genes to be "expressed" or "not expressed" and hence enables practitioners to robustly identify known cell populations using positive and negative markers.

As a second example, Lukassen et al (2020) sought to identify cells which are triple positive for proteins SARS-CoV-2 entry proteins ACE2, TMPRSS2, and FURIN. Due to dropout, only 3 of the 17,521 cells were ACE2+TMPRSS2+FURIN+, which significantly limited their analysis. Using ALRA, we identify 1,309 of these cells. In contrast, after imputation using methods that do not preserve biological zeros, such as MAGIC, a straightforward identification of these cells is not possible. While one could add a post-processing thresholding step to the output of other imputation methods, choosing a suitable threshold is far from trivial and in our limited experiments different thresholds can lead to dramatically different results.

Investigation on how reconstructed expression levels affect the various tSNE analysis adds little information of interest: it is highly likely even better performance would be achieved if not censoring some values to 0, or even if the tSNEs were applied to the principal components from the SVD/PCA rather than first reconstructing the matrix.

The reviewer suggests that the reconstruction and thresholding steps are inconsequential to the t-SNE visualizations. This is not the case; reconstruction and thresholding step denoises the matrix and improves the resulting visualization. This is particularly important when restricting the PCA to variable genes, as described below.

Consider the following two pipelines:

Pipeline A1: PCA \rightarrow t-SNE applied to first few leading PCs

Pipeline B1: ALRA (SVD \rightarrow Low-rank reconstruction \rightarrow Thresholding) \rightarrow PCA \rightarrow t-SNE applied to first few leading PCs

The reviewer suggests that ALRA's thresholding step is not necessary, and hence the SVD and low-rank reconstruction steps are redundant (because PCA is just SVD of the centered matrix). However, this is not the case. The thresholding step further denoises the matrix, based on the prior knowledge that a large number of values in the matrix are zero. In this context, we would like to point out that thresholding of small values to zero is a popular denoising operation that plays a key role in high dimensional statistics. In particular, for sparse high dimensional signals corrupted by noise, thresholding enjoys various optimality properties. Given the large sizes of scRNA-seq matrices, the thresholding step of ALRA should be viewed as a crucial denoising step, which has deep connections to the high-dimensional statistics literature.

The pipelines above do not include a step present in essentially every analysis: identification of variable genes. Therefore, as in other imputation methods papers, we include the variable genes selection step into the pipelines, as follows:

Pipeline A2: Identify variable genes \rightarrow PCA on variable genes \rightarrow t-SNE applied to first few leading PCs

Pipeline B2: ALRA (SVD \rightarrow Low-rank reconstruction \rightarrow Thresholding) \rightarrow Identify variable genes \rightarrow PCA on variable genes \rightarrow t-SNE applied to first few leading PCs

In the paper, we showed that this pipeline B2 results in dramatically improved results over pipeline A2. This is because the missing values in the variable genes have been imputed using information from the rest of the genes. Given that the subsequent PCA is restricted to the variable genes, this reconstruction step is essential to making those variable genes more informative. To summarize, the thresholding is a form of denoising and the reconstruction is essential because of the subsequent restriction of the analysis to variable genes.

Altogether, we emphasize that ALRA is a method for denoising scRNA-seq datasets which enables a wide variety of downstream analyses. ALRA is particularly unique in that it preserves biological zeros; this can be interpreted as setting to zero all expression values that are indistinguishable from noise. This may be viewed as an analog of gating in flow cytometry, where protein expression below a threshold is identified as noise and called "no expression." In the supplement, we provide a theoretical justification of ALRA's performance, which is the most rigorous analysis of a denoising method in the scRNA-seq literature (to the best of our knowledge). This theory, in combination with validation on over 10 real-life datasets, establishes ALRA as a useful tool for imputation of scRNA-seq datasets.

REVIEWER 2

In this paper, the authors developed a method called ALRA that imputes single-cell RNA-seq data using singular-value decomposition (SVD) plus thresholding. They demonstrated the performance of ALRA using three real datasets and three simulation studies (synthetic data), and they downsampled the original data to create the data that needed imputation and evaluated the imputation results by the original data. I believe this method would be useful for large-scale scRNA-seq datasets given its good computational efficiency, if the validation results were convincing. The authors showed that ALRA has improved accuracy measured by the Gini index, which benchmarks imputed scRNA-seq data against FISH data, over that of the existing imputation methods.

We would like to clarify that we demonstrate ALRA's performance on 11 real datasets, and a single simulation study. We summarize below:

- (1) Zheng et al. (2017): We show that ALRA preserves biological zeros more effectively than other methods. Biological zeros are defined using prior knowledge about marker gene expression (e.g. PAX5, NCAM1, etc.) and also by bulk RNA-sequencing (Hoeck et al. 2015). This result is in Figure 2a-c.
- (2) Stoeckius et al. (2015): ALRA preserves biological zeros, where zeros were defined using bulk RNA-seq and CITE-seq data
- (3) By preserving biological zeros, we are able to quantitatively compare the number of zeros expression marker genes between knockout and control in Gupta et al. (2019)
- (4) Tasic et al. (2018): We showed that ALRA more clearly separates known neocortical cell types than other methods, Hrvatin et al. (2018): Similarly, ALRA more clearly separates known mouse visual cortex cell types than other methods.
- (5) Torre et al. (2018): Imputation by ALRA results in gene expression distributions that are more consistent with the FISH of the same cell populations
- (6) We showed that ALRA better characterizes the distinct levels of Pdgfra expression known to exist in lamina propria cells from the colon of Pdgfra(EGFP/+) knock-in mice
- (7) Baron et al (2016), Chen et al. (2017), La Manno et al. (2016), Zeisel et al. (2015): ALRA improves separation of cell types from downsampled data in all four datasets, as mentioned by the reviewer
- (8) Simulation study using bulk RNA-seq from ImmGen, as mentioned by the reviewer

However, I have the following major concerns regarding the validity of the results. Most importantly, the authors need to provide more detailed figure captions to make the figures understandable.

We have expanded the figure captions to make the results more clear.

1. In the section "ALRA preserves biological zeros in simulated and real datasets," the authors defined biological zeros based on "genes with solid biological knowledge not expressed in a cell population." However, the authors did not provide literature or data evidence to justify their definitions, that is, what genes are not expressed in what cell types by what evidence.

We consider the following three separate approaches to define biological zeros:

- (1) Prior knowledge: We appreciate the reviewer pointing out that we did not cite literature to justify these definitions. We have added citations supporting our assumption that PAX5, NCAM1, and CD4 are specific to B cells, natural killer cells, and helper T cells respectively. For example, the Human Blood Atlas (<https://www.proteinatlas.org/humanproteome/blood>) is a recent project which determines genes specifically expressed in different blood cell types.

- (2) Bulk RNA-sequencing: We used bulk RNA-sequencing to identify genes which are not expressed in a cell population of interest. In particular, if a gene has 0 counts in a bulk RNA-seq sample of a homogenous cell population consisting of a large number of cells, we conclude that this gene is not expressed by that cell type. In doing so, we identified several hundred genes which should not be expressed in B cells, monocytes, T cells, and CD56 NK cells. We then showed that ALRA preserves these biological zeros in two scRNA-seq datasets of these populations better than competing methods (Figure 2C and Supplemental Table S2).
- (3) Knock-out experiment: In a dataset where the beta-catenin/Wnt pathway was conditionally knocked out, we show that ALRA allows for quantification of the proportion of cells with active Wnt.

2. In the analysis of "The importance of preserving biological zeros: an example", SAVER and scImpute were no longer compared. The authors should include these two methods to be consistent with other results.

The point of this section was not to benchmark methods, but rather to give a general example of why it is important to preserve biological zeros. Hence we did not include all the methods for comparison. In the revised version, we included the results for SAVER and scImpute as well.

3. It is unclear why ALRA favors diverse cell types. This important phenomenon needs to be justified.

In the presence of diverse cell types, there is a large variability in the columns of the matrix. In turn, this implies that the leading singular values are large. As is well known, and also evident from our mathematical analysis, the ability to accurately estimate the leading singular vectors and the rank-r approximation of the matrix depends on how large are the singular values and the spectral gap from the remaining smaller singular values. In the setting of few, similar cell types, the leading singular values are smaller. Consequently, the rank-r SVD of the observed matrix with many zeros is less accurate, which also affects the ability to accurately preserve biological zeros.

4. The choice of the $p = 0.001$ quantile seems arbitrary and unjustified. The choice of this threshold is crucial to the performance of ALRA, so the authors need to provide a practical guide for users to make the choice a data-driven way.

We thank the reviewer for pointing out that sensitivity to the threshold parameter p was not clear. Actually, the choice of p is almost completely inconsequential. In ALRA's original formulation, we used a threshold defined by the magnitude of the most negative value in each gene (i.e. $p=0$). In this case, there is no free parameter for choosing a threshold. In this later version we submitted to Nature Communications, we set $p = 0.001$ out of the theoretical concern that an outlying expression value could lead to too large of a cut-off, but we never had any empirical evidence of this occurring. We added a supplemental figure showing that the results of ALRA are very robust to the choice of threshold.

5. Many figures have vague captions and labels. For example, Figure S8 is hard to understand because the meaning of the y-axis is unclear. The message in Fig. S3 is also unclear.

We have clarified the captions and labels.

6. In the section “ALRA improves separation of cell types,” the authors claimed that “To quantitatively confirm that ALRA increases separation, we trained supervised random forests to classify the cells into cell types based on expression values”. However, the SAVER paper argued the other way: the cell-types labeled in Zeisel et al. were based on cell clusters found from the raw scRNA-seq data and not trustworthy, so imputation was needed. In fact, the SAVER paper used the cell clusters found from their imputed Zeisel et al. data instead of the cell types for evaluation. Given the argument in the SAVER paper, the use of cell types to train a supervised classification algorithm is unconvincing. The authors need to justify it.

Huang et al. identified clusters in the original dataset from Zeisel et al, and then downsampled the number of reads to produce an “observed” dataset. The clusters in the original dataset were considered to be “ground truth labels,” as they were determined using all available data. They then showed that applying SAVER to the downsampled dataset recovered these “ground truth” cluster labels. Specifically, they compared the clustering of each of the observed and imputed datasets to the “ground truth” dataset using the Jaccard index.

Similarly, we showed that ALRA recovers the “ground truth” cluster labels computed by Huang et al. even better than SAVER does. Instead of using Jaccard index, which requires clustering, we trained a random forest classifier to predict the cluster label and used the “out-of-bag-error” as a measure of accuracy. The advantage of this method is that it does not depend on re-clustering. For completeness, we also compared using Jaccard Index (see Table R1 below), and found similar results: ALRA substantially improves the Jaccard index in all cases, outperforming competing methods in 3 out of 4 datasets.

TABLE R1. Jaccard index between clusters on original dataset and clusters after imputing the downsampled dataset by various methods.

	Obs	SAVER	ALRA	DCA	MAGIC	scImpute
baron	0.50	0.80	0.86	0.84	0.50	0.38
chen	0.35	0.49	0.42	0.65	0.46	0.28
manno	0.36	0.47	0.47	0.47	0.46	0.32
zeisel	0.40	0.60	0.73	0.63	0.38	0.24

REVIEWER COMMENTS

Reviewer #1 (Remarks to the Author):

In this revision the authors have written a detailed response to concerns by the reviewers, and added some clarifications in the manuscript. The authors have also added some motivational examples for the utility of preserving zeros in analysis.

It is still not convincing the absolute zeros are more meaningful than e.g. multimodal distributions of count frequencies obtained by other methods. Yet the method is highly scalable and the underlying theoretical work is a substantial effort (though working under assumptions which seem quite divorced from the scRNA-seq measurement process).

Thanks to the clarifications made in the new manuscript it is now clear the authors are working on a scale of $\log(1,000 * x / s + 1)$, where x is a molecule count and s is a library depth. This is highly relevant for the authors method, as different choices in the arbitrary constants '1,000' and '1' in this scaling will affect the relation between zero and non-zero values. In particular this will affect how numbers are thresholded after performing the SVD reconstruction. The distribution of $\log(x / s + 1)$ is very different from $\log(1,000 * x / s + 1)$ which is very different from $\log(10,000 * x / s + 1)$. Not to mention that zeros can be shifted by changing the arbitrary offset '+ 1'. This issue is discussed in depth in Lun 2018 <https://www.biorxiv.org/content/10.1101/404962v1.full>.

The authors must demonstrate the effects of changing the scaling factors in the log normalization on the number of retained or imputed zeros. This will have substantial effect on e.g. the cell type gating analyses the authors use to motivate the method (such as the triple positive SARS receptor cells). If results are not robust to these arbitrary scaling factors, they will be difficult to trust.

Reviewer #2 (Remarks to the Author):

The authors have addressed most of my previous comments. I only have two remaining comments for this version.

1. In the authors' response to my question about why two methods, SAVER and scImpute, were left out in the section "The importance of preserving biological zeros," they added the following sentence:

"SAVER's results falsely suggest that the Wnt activity in the knockout and control is identical, whereas scImpute falsely suggests that DC differentiation is similar between knockout and control."

However, there is no evidence to support this statement in Figure S3, where the authors compared ALRA with DCA and MAGIC.

2. I still don't understand the authors' explanation about why ALRA performs better when cell types are more diverse. The authors wrote

“In the presence of diverse cell types, there is a large variability in the columns of the matrix. In turn, this implies that the leading singular values are large.”

To me, there is a gap between the greater diversity of cell types and the larger leading singular values. I think the primary reason for my lack of understanding is the mathematical definition of "cell type diversity." If the authors can write down the mathematical derivation to support this statement, it will clarify my confusion.

RESPONSE TO THE REVIEWERS' COMMENTS

REVIEWER 1

In this revision the authors have written a detailed response to concerns by the reviewers, and added some clarifications in the manuscript. The authors have also added some motivational examples for the utility of preserving zeros in analysis.

It is still not convincing the absolute zeros are more meaningful than e.g. multimodal distributions of count frequencies obtained by other methods. Yet the method is highly scalable and the underlying theoretical work is a substantial effort (though working under assumptions which seem quite divorced from the scRNA-seq measurement process).

We thank the reviewer for this summary of our changes, and especially for noting the value of our theoretical work.

Our perturbation analysis was done under both a dropout model as well as a more realistic multinomial model. We believe the reviewer's comments about the assumptions being "divorced from the scRNA-seq measurement process" are referring to the dropout model, which has recently fallen out of favor in the scRNA-seq community. This set of assumptions has largely been supplanted by more realistic models where the excess of zeros are explained by insufficient sampling of reads from each cell [1], such as the multinomial model in [2].

In the revised supplementary, we added further details, including a new lemma, explaining why our Theorem is valid under the multinomial model. We also added an introduction section which frames the analysis and makes it clear that we extend our results to the multinomial setting. We appreciate that the reviewer helped us see that the analysis under this more realistic model should be expanded and featured more prominently.

Beyond our assumptions about the measurement process itself, our analysis also requires assumptions about the spectral structure of the expression matrix. Most importantly, we assume that the unobserved, underlying expression matrix is low-rank. The biological justification for this assumption is that genes do not act independently, but rather as part of concerted gene modules, resulting in a low-rank matrix. This is the same assumption that underlies the use of principal component analysis as a preprocessing step, as is standard in many analysis pipelines. We make further assumptions that are common in the low-rank matrix completion literature, and we list them in Section 2 of our theoretical analysis. In this revision, we expand upon why these assumptions are reasonable in the specific setting of scRNA-seq data.

*Thanks to the clarifications made in the new manuscript it is now clear the authors are working on a scale of $\log(1,000 * x / s + 1)$, where x is a molecule count and s is a library depth. This is highly relevant for the authors method, as different choices in the arbitrary constants '1,000' and '1' in this scaling will affect the relation between zero and non-zero values. In particular this will affect how numbers are thresholded after performing the SVD reconstruction. The distribution of $\log(x / s + 1)$ is very different from $\log(1,000 * x / s + 1)$ which is very different from $(\log(10,000 * x / s + 1))$. Not to mention that zeros can be shifted by changing the arbitrary offset '+ 1'. This issue is discussed in depth in Lun 2018 <https://www.biorxiv.org/content/10.1101/404962v1.full>.*

The authors must demonstrate the effects of changing the scaling factors in the log normalization on the number of retained or imputed zeros. This will have substantial effect on e.g. the cell type gating analyses the authors use to motivate the method (such as the triple positive SARS receptor cells). If results are not robust to these arbitrary scaling factors, they will be difficult to trust.

We thank the reviewer for pointing out this important literature on the effect of scaling factors on downstream analysis. The reviewer raises the noteworthy concern that these scaling factors may also affect ALRA's imputation results, particularly with respect to preserving zeros. We added two analyses which show that ALRA is not sensitive to the choice of scaling factors. In Table S6, we show that the rates of biological zero preservation (ZP) and total zeros completed (TC) in the peripheral blood mononuclear cells from Figure 2B do not change appreciably with different choices of scaling factor. In Table S7, we show that different choices of scaling factor do mildly affect the rank estimation, but still result in generally consistent numbers of triple positive SARS receptor cells (as in Table S4, ALRA is not sensitive to the estimated rank of the matrix). In each of these examples, we varied the scaling factor from 1,000 to 20,000. This choice reflects the two approaches we have seen used most commonly in the literature: 1) to simply use 10,000, which is the default in packages like Seurat and 2) to use the median UMI count in the dataset, after filtering. For example, the median UMI counts in three datasets we analyzed in this paper, Zheng et al, Gupta et al., and Lukassen et al., were 1,426, 8,194, and 4,436—well within our chosen range.

The ideal approach to normalizing scRNA-seq expression data is an active area of research. ALRA's results are based on the most popular approach, which is the $\log(10,000 \times +1)$ transformation. As the above examples demonstrate, the choice of this scale factor does not appreciably change the imputation results.

REVIEWER 2

The authors have addressed most of my previous comments. I only have two remaining comments for this version.

1. In the authors' response to my question about why two methods, SAVER and scImpute, were left out in the section "The importance of preserving biological zeros," they added the following sentence:

"SAVER's results falsely suggest that the Wnt activity in the knockout and control is identical, whereas scImpute falsely suggests that DC differentiation is similar between knockout and control."

However, there is no evidence to support this statement in Figure S3, where the authors compared ALRA with DCA and MAGIC.

We thank the reviewer for pointing out that we had omitted the reference to Table S1 which supports this statement. In Table S1, we see that the markers for Wnt activity (AXIN2+ & LEF1+) are essentially identical in the mutant (76%) and wildtype (77%) following SAVER, when actually the percent should be much smaller in the mutant. We have added a reference to Table S1 following that statement, and we also added plots for scImpute and SAVER to Figure S3.

2. I still don't understand the authors' explanation about why ALRA performs better when cell types are more diverse. The authors wrote

"In the presence of diverse cell types, there is a large variability in the columns of the matrix. In turn, this implies that the leading singular values are large."

To me, there is a gap between the greater diversity of cell types and the larger leading singular values. I think the primary reason for my lack of understanding is the mathematical definition of "cell type diversity." If the authors can write down the mathematical derivation to support this statement, it will clarify my confusion.

We thank the reviewer for this comment. In what follows, we explain the mathematical foundations of this statement. Consider first a data matrix which consists of a single cell type. Denote by X the underlying expression matrix before the multinomial sampling. As individual cells from the same type still have some small within-class variability, the matrix X is approximately low rank but with relatively small spread, hence small singular values. In this context, recall that the square of the singular values, which are the eigenvalues of the (non-centered) covariance matrix, capture the variability of the columns of X , which indeed is rather small within a single cell type. Applying ALRA to the corresponding observed data, the leading eigenvectors attempt to capture this relatively small within-class variability and are more affected by the noise.

In contrast, consider a dataset consisting of K different cell types. Here, the small within-class variability characteristic of each cell type is still present, but added to it is a significantly larger variability in the directions between the mean column vectors of the different cell types. Namely, in a generic case, the underlying expression matrix X will have $K - 1$ large singular values, which would be much larger than those of a matrix discussed previously with a single cell type. In this case, applying ALRA to the multinomial sampled data, the SVD step will capture the between class variability, and would be far more resilient to noise. Furthermore, these leading eigenvectors often capture the marker genes that are specific to the various cell types, thus allowing ALRA to better impute and preserve them. This argument applies to any type of structure or "signal" in the data; for simplicity, we used K distinct cell types in this explanation. However, non-clustered data with large biological variation would similarly have larger singular values compared to a dataset with only technical noise.

REFERENCES

- [1] Valentine Svensson. Droplet scRNA-seq is not zero-inflated. *Nature Biotechnology* 38(2)147-150. 2020.

- [2] William F. Townes, Stephanie C. Hicks, Martin J. Aryee, and Rafael A. Irizarry. Feature selection and dimension reduction for single-cell RNA-Seq based on a multinomial model. *Genome Biology* 20(1)1-16. 2019.

REVIEWER COMMENTS

Reviewer #1 (Remarks to the Author):

Review

In this revision, Linderman et al have changed some phrasing in the manuscript, and slightly extended descriptions in the supplementary notes. The authors have added a comparison of the number of called zeros in a couple of their applications depending on scaling factors in their data transformation step. This analysis does not address the issues raised in the previous revision.

In the theoretical supplemental notes the authors have expanded the section on how the theory applies to multinomial data. In practice however the authors never apply their algorithm to data following either of their two assumed distributions (spike-and-slab normal and multinomial). The authors make use of $\log(1000 * x / s + 1)$ scaled counts, which does not follow either of those assumed distributions. As such, the theoretical work does not apply to the data used in practice.

As discussed in the previous review, changing the arbitrary scaling factors 1000 and 1 will change the distribution properties of the expression matrix. As such these will greatly impact the meaning of the width of the distribution around 0 in the reconstruction. It is also not clear how changing the arbitrary 1, which moves the untransformed '0' in the transformed distribution. The paper linked in the previous review points out how the multiplicative scaling factor interacts with the additive offset, and these need to be considered together.

The brief tests on how results from the PBMC and bronchial epithelial data are changed due to different scaling factors does not sufficiently address the underlying issue. How does contracting and expanding the distribution around 0 by changing the scaling factor alter the distribution of reconstructed values? How does this affect how 'close' the non-zero distribution is to the noise distribution symmetrically distributed around 0?

Regarding the extended discussion on the multinomial case, it would be good to show at what threshold of probability p_{ij} the ALRA method will consider it indistinguishable from 0. In practical scRNA-seq datasets many genes known to be functionally present, such as transcription factors, have p_{ij} values as low as $1e-5$ (roughly corresponding to single digit molecule numbers in individual cells, not even considering the ~10% efficiency of scRNA-seq counting).

Since the authors have investigated the mathematics of the multinomial count distribution, they could also demonstrate the method on the multinomial count data, and contrast those results with the transformed results.

In the added supplementary tables the authors state expression is of the form $\log(\alpha * x + 1)$ where α is one of 1000, 5000, 10000 etc. But in the methods section the authors write they use $\log(1000 * x + 1)$.

$x / s + 1$). These different transformations lead to very different distributions. What was actually investigated?

Reviewer #2 (Remarks to the Author):

The authors use the singular value decomposition (SVD), a classic matrix method, combined with thresholding on the SVD reconstructed values to perform scRNA-seq data imputation. That is, the thresholded SVD reconstructed values are the “imputed” values. This method ALRA has a main contribution: introducing a new way to identify biological zeros.

To help convince me the validity of the identification of biological zeros and the quality of the imputed value, I would like to see the authors address the following issues:

1. The theoretical derivation in this paper suggests that biological zeros follow Gaussian distributions with mean zero after SVD reconstruction. However, in Figure 2A, for gene CD8A, the distributions of the reconstructed gene expressions are quite different in four cell types. Hence, it is not obvious how the theoretical results are applicable when the cell types are unknown, because it is likely that non-biological zeros in one cell type will be thresholded as zeros if the threshold is not cell-type-specific.
2. Related to 1, the simulation using bulk RNA-seq of five T cell populations from the Imm-Gen consortium doesn't reflect the cell type differences. A more realistic simulation should generate scRNA-seq data from a mixture of distinct cell types.
3. Besides, according to the theoretical results, the reconstructed value of each biological zero count is sampled from a zero-mean Gaussian distribution by itself. However, it is not guaranteed that the reconstructed values of biological zeros are independently and identically distributed as one zero-mean Gaussian distribution.
4. Given that this algorithm will threshold all the values below the set threshold, the authors should show the percentage of non-zero values in the original matrix that has been thresholded. The authors should also report the ratio between the number of negative and positive reconstructed values that were thresholded to zero for every dataset.
5. Based on the description in page 11, the ALRA algorithm will change all the values after imputation, no matter if they are zeros or not. Although a rescaling process has been proposed, the authors should also show the correlations between imputed non-zero values and original non-zero values among all the genes.
6. The authors only show the distribution of one gene before and after imputation in S15, S16, S17. They need to show the distribution before and after imputation for more genes using ALRA. Besides, the

authors should show the following three distributions of the same gene. 1. The distribution of this gene before imputation. 2. The distribution of this gene after SVD reconstruction before thresholding. 3. The distribution of this gene after thresholding and rescaling.

7. In the real data analysis, the authors run the analysis on UMI data. They do not discuss the performance of this method on non-UMI data, such as Smart-seq2 data.

8. The authors need to explain the low rank assumption and why it is biologically meaningful. If the truth is assumed to be low rank, does it mean that cells in the same type/subtype are biologically identical? I think this is key for biologists.

RESPONSE TO THE REVIEWERS' COMMENTS

We thank the reviewers for their careful reading of our manuscript and the helpful suggestions. We believe the paper has been substantially improved through this process. Below we provide point-by-point responses to the comments and suggestions raised in the third round of review.

REVIEWER 1

In this revision, Linderman et al have changed some phrasing in the manuscript, and slightly extended descriptions in the supplementary notes. The authors have added a comparison of the number of called zeros in a couple of their applications depending on scaling factors in their data transformation step. This analysis does not address the issues raised in the previous revision.

In the second round of review, the reviewer expressed concern that ALRA's performance at preserving biological zeros would be affected by the choice of scaling factors in the data transformation step. We showed in two datasets that the scaling factors did not substantially change the results. In this revision, we show this more extensively (see below).

*In the theoretical supplemental notes the authors have expanded the section on how the theory applies to multinomial data. In practice however the authors never apply their algorithm to data following either of their two assumed distributions (spike-and-slab normal and multinomial). The authors make use of $\log(1000 * x / s + 1)$ scaled counts, which does not follow either of those assumed distributions. As such, the theoretical work does not apply to the data used in practice.*

4/6 Follow-Up: The authors don't need to prove mathematically the transformed distributions satisfy their theorems. While the mathematical supplement is interesting, I give it a fairly low weight in judging the method, because as the authors point out in their response, the theoretical models lack a lot of aspects of observed data.

As explained in our inquiry concerning this review (communicated by the editor to this reviewer on 4/1), ALRA's empirical performance in detecting biological zeros is not affected by the log transform. That is, applying ALRA to the column-normalized data (i.e. dividing the count data vector of each cell by the sum of its entries, without log transform) and to the log-transformed data yield similarly high rates of biological zero preservation (Tables 1-3 in the Appendix). However, pre-processing with the log transform prior to imputation with ALRA substantially improves the performance of downstream analyses like t-SNE. In this revised version of the paper, we have added a section titled "Variance Stabilization" to the theoretical analysis explaining the effect of the log transform and demonstrating why it is helpful in this setting. Finally, we emphasize that the main contribution of the theoretical analysis is to provide intuition and rationale for the symmetry of biological zeros—and we consistently observe the same symmetry using the log-transformed data.

As discussed in the previous review, changing the arbitrary scaling factors 1000 and 1 will change the distribution properties of the expression matrix. As such these will greatly impact the meaning of the width of the distribution around 0 in the reconstruction. It is also not clear how changing the arbitrary 1, which moves the untransformed '0' in the transformed distribution. The paper linked in the previous review points out how the multiplicative scaling factor interacts with the additive offset, and these need to be considered together.

The brief tests on how results from the PBMC and bronchial epithelial data are changed due to different scaling factors does not sufficiently address the underlying issue. How does contracting and expanding the distribution around 0 by changing the scaling factor alter the distribution of reconstructed values? How does this affect how 'close' the non-zero distribution is to the noise distribution symmetrically distributed around 0?

*4/6 Follow-up: I see the authors point that it is common in the field to simply perform $\log(x + 1)$ scaling and that is considered a standard. But as have been pointed out by several papers (the Lun paper linked in an earlier review, and more recently in <https://doi.org/10.1093/bioinformatics/btab085>), the arbitrary coefficients used when scaling the data will specifically affect how close zero-values and non-zero values are to each other on the transformed scale. And this closeness is exactly what the authors' method uses to determine 'biological zeros'. I believe that if I use ALRA, and change $\log(x / s + 1)$ to e.g. $\log(100 * x / s + 0.001)$, or some other combination of arbitrary scaling factors, I can make ALRA as many or as few 'biological zeros' as I want. This is why I want the authors to demonstrate how data distribution shifts depending on transformation, and how the reconstructed values are also distributed depending on transform.*

We apply the standard pre-processing transformation $y_{ij} = \log(\alpha \cdot x_{ij}/s + c)$ to the expression value x_{ij} of the i th cell and j th gene, where α is a scaling factor, c is an offset, and s is the library size. The reviewer raised an important concern: how do choices of α and c affect ALRA's performance? In particular, will different choices of these parameters affect the number of biological zeros?

We first note that while α is completely arbitrary, the offset c is not. If we intend to preserve biological zeros, the only possible choice of c is 1, because $\log(c) = 0$ only for $c = 1$. Any other choice would shift all the zeros to a non-zero value and hence would be inconsistent with the goal of preserving biological zeros.

In contrast, users may reasonably choose different values for the scaling factor α . In this revision, we show that the number of biological zeros preserved does not substantially change based on the choice of α in four datasets (Tables S6-S9). We also show how the distribution of expression values changes with the choice of scaling factor in Supplemental Figure S18. As this figure shows, the shape of the resulting distribution is very robust to the choice of scaling factor. In turn, this implies that the preservation of zeros is nearly unaffected by the value of the scaling factor. We conclude that the scaling factor α does not actually "affect how close the zero-values and non-zero values are to each other on the transformed scale."

In summary, these results establish that the reviewer's concern that "[by changing scaling factors], I can make ALRA as many or as few 'biological zeros' as I want" is not realized.

Regarding the extended discussion on the multinomial case, it would be good to show at what threshold of probability p_{ij} the ALRA method will consider it indistinguishable from 0. In practical scRNA-seq datasets many genes known to be functionally present, such as transcription factors, have p_{ij} values as low as $1e-5$ (roughly corresponding to single digit molecule numbers in individual cells, not even considering the 10% efficiency of scRNA-seq counting).

We have added a simulation to assess the probability of imputing a technical zero as a function of p_{ij} . We simulated 5,000 cells from the bulk RNA-seq profiles of five T-cell populations using the multinomial model described in the Online Methods section titled, "Simulated scRNA-seq data." The number of reads sampled from each cell was randomly chosen from the read counts in the purified PBMC dataset, so that the distribution would be realistic. After running ALRA, we then plotted the probability that each technical zero was completed as a function of the probability of expression, p_{ij} . In this simulation (Supplemental Figure 11), we see that at around $p_{ij} = 1e-5$, $\sim 75\%$ of the technical zeros are imputed, and at around $p_{ij} = 1e-4$, nearly 100% of the technical zeros are imputed.

Since the authors have investigated the mathematics of the multinomial count distribution, they could also demonstrate the method on the multinomial count data, and contrast those results with the transformed results.

We originally assessed ALRA's performance on data that was log transformed after it was generated from a multinomial model. We repeated the simulation without the log transform (i.e. after dividing the count data vector of each cell by the sum of its entries, without log transform), as requested by the reviewer. Specifically, ALRA's performance in preserving biological zeros without preprocessing by the log-transform is shown using this simulated data and two real-life datasets in Tables 1-3 of the Appendix (i.e. the theoretical analysis).

*In the added supplementary tables the authors state expression is of the form $\log(\alpha * x + 1)$ where α is one of 1000, 5000, 10000 etc. But in the methods section the authors write they use $\log(1000 * x / s + 1)$. These different transformations lead to very different distributions. What was actually investigated?*

*4/6 Follow-up: On a related note, the authors responded they use $\log(x + \alpha)$ in all calculations. This means all observed expression levels are confounded by library size which is known to be the largest explaining factor for 'zeros' in scRNA-seq data. The learned low-rank representations are likely mostly picking up the signal of library size rather than biological variation, and it would be hard to interpret the results. If the authors still want to use $\log(x + \alpha)$ rather than $\log(\beta * x / s + \alpha)$ they need to demonstrate their inferred biological zeros do not depend on library size.*

In our response, we misunderstood the reviewer's notation and stated we used $\log(\alpha \cdot x + 1)$, when in reality we do divide by library size in all experiments. We do this for the exact reason stated by the reviewer: to remove the effect of library size in the resulting analysis.

To be perfectly clear, the transformed expression of gene j in cell i is

$$y_{ij} = \log\left(\frac{\alpha}{\sum_j x_{ij}} \cdot x_{ij} + c\right).$$

That is, we divide by the library size, multiply by an arbitrary scaling factor of $\alpha = 10,000$, add the "pseudo-count" of $c = 1$, and take the log. We have clarified this transformation in the text as well. As previously mentioned, we also show that the preservation of biological zeros does not substantially change with different values of α .

REVIEWER 2

The authors use the singular value decomposition (SVD), a classic matrix method, combined with thresholding on the SVD reconstructed values to perform scRNA-seq data imputation. That is, the thresholded SVD reconstructed values are the “imputed” values. This method ALRA has a main contribution: introducing a new way to identify biological zeros.

To help convince me the validity of the identification of biological zeros and the quality of the imputed value, I would like to see the authors address the following issues:

1. The theoretical derivation in this paper suggests that biological zeros follow Gaussian distributions with mean zero after SVD reconstruction. However, in Figure 2A, for gene CD8A, the distributions of the reconstructed gene expressions are quite different in four cell types. Hence, it is not obvious how the theoretical results are applicable when the cell types are unknown, because it is likely that non-biological zeros in one cell type will be thresholded as zeros if the threshold is not cell-type-specific.

The reviewer makes the important point that the threshold is not cell-type specific. We do not require the user to pre-specify cell types, which is especially important as cell-type identification is very much an open problem in scRNA-seq data analysis. In this respect, ALRA’s simplicity is a strength, allowing it to be applied to datasets with and without clearly defined cell-types.

The gene-specific threshold which we use to restore the biological zeros is derived from the most negative imputed values for each gene. As the reviewer points out in Figure 2A, the threshold for CD8A is determined by the CD4 cells, which have a wider distribution, resulting in a larger threshold. This will indeed result in imputing fewer technical zeros, as they cannot be distinguished from the biological zeros using our approach.

This example nicely demonstrates why we make no claims about imputing technical zeros. Some will inevitably be below the threshold, and hence kept as zeros. This is why we consider ALRA to be a conservative method for imputation: zeros are only imputed if we have strong evidence that they are not biological zeros. In spite of this, we do find that ALRA also correctly imputes a large proportion of technical zeros, as demonstrated throughout the paper. Furthermore, based on the reviewer’s suggestion below, we added a measure of “excess zeros,” which estimates the number of technical zeros that were not completed in each gene (section “Excess Zeros” under Online Methods).

Finally, we note that a more refined imputation (e.g. based on cell types or some other more “local” approach to thresholding) is an interesting topic for further research, but beyond the scope of this manuscript.

2. Related to 1, the simulation using bulk RNA-seq of five T cell populations from the Imm-Gen consortium doesn’t reflect the cell type differences. A more realistic simulation should generate scRNA-seq data from a mixture of distinct cell types.

We thank the referee for this comment. We agree that a more realistic simulation would contain different cell types. In the 11 real-world (i.e. non-simulated) datasets used to benchmark ALRA throughout the paper, many of them are extremely diverse. For example, the Zheng et al. and Stoeckius et al. datasets contain celltypes as distinct as B cells, monocytes, and T cells. However, in this specific simulation, our goal is to demonstrate ALRA’s ability to preserve biological zeros in a challenging setting. To this end, we designed the simulation with sub-populations of T cells because it is more difficult to preserve biological zeros in datasets with relatively small differences between clusters (i.e. subpopulations of T cells) than if the clusters are dramatically different (i.e. distinct cell types, like B cells and T cells). Any imputation is effectively a type of smoothing, and imputation methods tend to oversmooth subtle differences between cell types. By using cell populations with more subtle differences in this simulation, we show that ALRA does not oversmooth. In contrast, other methods falsely impute biological zeros for genes in a given population that does not express them, simply because a similar population expresses them strongly.

As noted by the reviewer above, there is a trade-off between preservation of biological zeros and imputation of technical zeros. With highly distinct cell types, in general the preservation of biological zeros is easier, at the expense of a lower completion rate for technical zeros. In Table 1 of this response we expanded the simulation to include 8 cell types, including dendritic cells, neutrophils, B cells, and T cells. ALRA was run on the data matrix combining all these cell types. Compared to the simulation including 5 cell types, the rates of preserving biological zeros were higher, while the technical zero completion rates were lower.

3. Besides, according to the theoretical results, the reconstructed value of each biological zero count is sampled from a zero-mean Gaussian distribution by itself. However, it is not guaranteed that the reconstructed values of biological zeros are independently and identically distributed as one zero-mean Gaussian distribution.

We showed that under the stated assumptions, the reconstructed value of each biological zeros is symmetric around zero. To justify ALRA's procedure, we do not need the values to be identically distributed, as mentioned by the reviewer—this symmetry is sufficient. However, the reviewer is correct that we do require that the values not be strongly dependent. This is difficult to show theoretically, but we demonstrate using simulations that the values are nearly independent. We discuss this issue at length in Remark 1 following the theorem and the section titled Simulation at the end of the theoretical analysis.

4. Given that this algorithm will threshold all the values below the set threshold, the authors should show the percentage of non-zero values in the original matrix that has been thresholded.

In the ALRA algorithm, originally non-zero values are restored to their original value following the thresholding. That is, no entry that was non-zero prior to ALRA will be set to zero after ALRA. This is an important point that we have made clear in the revised manuscript.

The authors should also report the ratio between the number of negative and positive reconstructed values that were thresholded to zero for every dataset.

We thank the reviewer for suggesting we investigate this ratio—it is an informative measure that we had not previously considered. Given that the biological zeros are symmetrically distributed around zero, asymmetry in this distribution could be due to technical zeros. After experimenting with this idea, we propose a similar measure that is essentially the same as the ratio proposed by the reviewer, but which we believe may be slightly more interpretable.

For a gene i with threshold τ_i , define the number of excess zeros as

$$\psi_i = |\{j : 0 < y_{ij} < \tau_i \text{ and } x_{ij} = 0\}| - |\{j : y_{ij} < 0 \text{ and } x_{ij} = 0\}|,$$

where x_{ij} and y_{ij} denote the expression values before and after low-rank approximation, respectively, for the i th gene and j th cell. That is, the number of positive values set to zero in excess of the number of negative values set to zero. If there are no technical zeros below the threshold, then the number of positive values below the threshold (first term) will be nearly the same as the number of negative values (second term), and hence ψ_i will be close to zero. Conversely, if there are more positive values below the threshold than negative values, this suggests that a large number of technical zeros are also below the threshold will be set back to zero, as they cannot be distinguished from the biological zeros. To demonstrate this measure, we plot 6 genes with low ψ and 6 genes with high ψ (Supplemental Figure 20 and 21).

Of note, the average ψ_i across all genes in the Hrvatin et al. dataset is 11,563 (about 17% of the number of cells), but we believe it is more useful to use on a gene-by-gene basis. We have now implemented this measure as a function in the ALRA package, and we encourage users to check the number of excess zeros in genes of interest post-ALRA. We again thank the reviewer for this suggestion, which we believe ALRA users will find it to be an informative measure of how many technical zeros were not completed in specific genes of interest.

5. *Based on the description in page 11, the ALRA algorithm will change all the values after imputation, no matter if they are zeros or not. Although a rescaling process has been proposed, the authors should also show the correlations between imputed non-zero values and original non-zero values among all the genes.*

The heart of the ALRA procedure is singular value decomposition (SVD), which can be viewed as a denoising method [2–5]. Hence, ALRA not only imputes technical zeros, but the rank- r approximation can also yield estimates of the non-zero values.

For example, under the assumption that the observed matrix is low-rank corrupted by additive noise, the values obtained from the low-rank approximation are closer to the truth. From this perspective, we would like to emphasize that neither a high nor low correlation value would reflect ALRA's performance. However, we recognize that users may want to know how much the non-zero values have changed. The correlation may be an intuitive measure of how much the non-zero values were denoised. For example, in the Hrvatin et al dataset this value is 0.23, whereas in the purified PBMCs from Zheng et al. the correlation is 0.56.

We have updated the manuscript to emphasize that the non-zero values are also denoised, and we include the correlations for these two datasets as examples.

6. *The authors only show the distribution of one gene before and after imputation in S15, S16, S17. They need to show the distribution before and after imputation for more genes using ALRA. Besides, the authors should show the following three distributions of the same gene. 1. The distribution of this gene before imputation. 2. The distribution of this gene after SVD reconstruction before thresholding. 3. The distribution of this gene after thresholding and rescaling.*

We have added these three distributions requested by the reviewer for 12 genes from the Hrvatin et al. dataset. We chose genes that demonstrate the excess zeros measure that was inspired by this reviewer's comments in (4) above. Specifically, we chose (at random) six of the most variable genes that were in the top quartile of excess zeros, and six of the most variable genes that were in the bottom quartile of excess zeros (Supplemental Figures 20 and 21).

7. *In the real data analysis, the authors run the analysis on UMI data. They do not discuss the performance of this method on non-UMI data, such as Smart-seq2 data.*

We have revised the manuscript to clarify that ALRA is intended for use with UMI data. This is consistent with other imputation methods, such as SAVER. We thank the reviewer for pointing out that this was not clear in the original manuscript.

8. *The authors need to explain the low rank assumption and why it is biologically meaningful. If the truth is assumed to be low rank, does it mean that cells in the same type/subtype are biologically identical? I think this is key for biologists.*

The low-rank assumption is at the heart of our proposed algorithm, and we thank the reviewer for the opportunity to explain it in greater detail. The biological justification for this assumption is that the genes do not act independently, but rather in concert, forming groups of highly correlated genes referred to as gene modules. Correlation between the features (genes) necessarily means that the eigenvalues of the covariance matrix will decay rapidly, making it possible to model the true expression matrix as a low-rank matrix. This is the reason why principal component analysis is so effective that it is used in nearly every scRNA-seq pipeline.

Crucially, this assumption does **not** require that the cells in each type/subtype are identical. We call the true expression matrix A consisting of m cells (rows) vs. n genes (columns) low-rank if we can write it as

$$A = \sum_{i=1}^k u_i \sigma_i v_i^T,$$

where u_i are length m vectors, v_i are length n vectors, σ_i are real numbers, and k is much smaller than m and n . We think of each v_i as representing a “gene module,” and each u_i representing the contribution of each gene module to each cell. That is, the c th cell can be written as

$$\sum_{i=1}^k u_i[c] \sigma_i v_i^T,$$

where $u_i[c]$ denotes the c th index of the vector u_i . This means that each cell is a weighted combination of the gene programs. Cells of the same type may have a similar combination of gene programs, but they are not at all assumed to be identical. Put differently, the entries of u_i corresponding to cells of the same type are not assumed to be identical, as the strength of each gene program will vary across cells, even within the same cell type.

In addition to the biological justification of genes functioning in correlated gene modules, there is also a practical reason why low-rank approximation is appropriate. For simplicity, suppose the true matrix A is corrupted by a noise matrix E ,

$$\hat{A} = A + E.$$

The only singular values/vectors of A that can be recovered are those that are larger than the largest singular value of E , i.e. the operator norm of E . Even if the true expression matrix was not low-rank, the only signal that can be recovered from \hat{A} is actually a low-rank matrix. The “rank-estimation” procedure we proposed in this paper seeks to estimate exactly this threshold where signal ends and noise begins. In that sense, taking the low-rank approximation is justified even when the true matrix is not exactly low rank, as removing singular vectors/values below the noise threshold yields a better estimate of the true matrix. We refer the reviewer to recent work by our group characterizing the noise threshold in count data, including scRNA-seq [6].

In summary, the low-rank assumption is biologically justified by the intricate correlation structure of gene expression. Furthermore, even if the true rank is much higher, recovery of a low-rank matrix is the best one can do in the presence of noise.

TABLE R1. Proportion of biological zeros preserved (ZP) and technical zeros completed (TC) after running ALRA on a simulated scRNA-seq dataset with 8 distinct cell types.

	ZP	TC
Dendritic Cells	0.99	0.63
Neutrophils	1.00	0.50
T-reg Cells	0.99	0.57
CD4 TConv Cells	0.98	0.59
CD8 T Cells	0.99	0.58
B Cells	0.99	0.59
gdT Cells	0.98	0.71
NK T Cells	0.99	0.65

REFERENCES

- [1] Takashi Ohkawa, Shuhji Seki, Hiroshi Dobashi, Yuji Koike, Yoshiko Habu, Katsunori Ami, Hoshio Hiraide, and Isao Sekine. Systematic characterization of human cd8+ t cells with natural killer cell markers in comparison with natural killer cells and normal cd8+ t cells. *Immunology*, 103(3):281–290, 2001.
- [2] Orly Alter, Patrick O Brown, and David Botstein. Singular value decomposition for genome-wide expression data processing and modeling. *Proceedings of the National Academy of Sciences*, 97(18):10101–10106, 2000.
- [3] Matan Gavish and David L Donoho. The optimal hard threshold for singular values is. *IEEE Transactions on Information Theory*, 60(8):5040–5053, 2014.
- [4] Qiang Guo, Caiming Zhang, Yunfeng Zhang, and Hui Liu. An efficient svd-based method for image denoising. *IEEE transactions on Circuits and Systems for Video Technology*, 26(5):868–880, 2015.
- [5] Mansour Nejati, Shadrokh Samavi, Harm Derksen, and Kayvan Najarian. Denoising by low-rank and sparse representations. *Journal of Visual Communication and Image Representation*, 36:28–39, 2016.
- [6] Boris Landa, Thomas TCK Zhang, and Yuval Kluger. Biwhitening reveals the rank of a count matrix. *arXiv preprint arXiv:2103.13840*, 2021.

REVIEWER COMMENTS

Reviewer #1 (Remarks to the Author):

With this revision the authors have clarified important details from the previous revision. The authors have added figures which demonstrate how results are affected by user choices in and data characteristics for several genes.

No further issues.

Reviewer #2 (Remarks to the Author):

The authors have addressed some of the two reviewers' previous comments. However, the authors did not highlight the corresponding changes in the manuscript; hence, I could not tell whether the authors revised the manuscript or not. I felt that both reviewers' comments are critical issues that readers may raise, so it is necessary for the authors to revise the manuscript accordingly. Regarding this revised manuscript, I still have the following major concerns about the reproducibility of results and the quality of the imputed values by ALRA.

1. In Figure 3A, the authors compared ALRA with several other imputation methods. However, the authors did not list the parameters used for the other imputation methods, and neither did they release the code for reproducing the results (the GitHub code available at <https://github.com/KlugerLab/ALRA-paper> was from three years ago and outdated). The code must be up-to-date so that all the results in the paper are reproducible.
2. The authors showed that ALRA is better at preserving biological zeros in a dataset with only four cell types (Figure 2). In another analysis, they showed that ALRA is better at separating cell types when there are diverse cell types (Figure 3). The two conclusions are somehow contradictory: does ALRA work better with fewer cell types or more cell types? The authors need to make comparisons using the same metrics under the two settings (few cell types vs. multiple cell types).
3. The distributions the authors plotted in Fig. S20 / S21 are worrisome, suggesting that ALRA cannot preserve the dynamic range of non-zero expression values before imputation. For example, among the three plots in the fourth row in Fig. S21, the leftmost plot shows a range of 1-2, but after imputation, the range expands to 1-5. Can authors justify these results?
4. The correlations between the non-zero values before and imputation are very low: 0.23 - 0.56. This low correlation may indicate that ALRA's model assumption is not valid. Also, given that the non-zero expression values are altered greatly by ALRA, how should users interpret or use the imputed expression values output by ALRA?

5. Regarding the low-rank assumption, can the authors show some of the “gene modules” they have described in the response? For example, does a gene module (genes with large weights in a vector v_i) correspond to a set of genes that are highly correlated in a certain cell type? Why is this gene module biologically meaningful?

RESPONSE TO THE REVIEWERS' COMMENTS

We thank the reviewers for their careful reading of our manuscript and the helpful suggestions. We believe the paper has been substantially improved through this process. Below we provide point-by-point responses to the comments and suggestions raised in this fourth round of review.

REVIEWER 1

With this revision the authors have clarified important details from the previous revision. The authors have added figures which demonstrate how results are affected by user choices in and data characteristics for several genes.

No further issues.

REVIEWER 2

The authors have addressed some of the two reviewers' previous comments. However, the authors did not highlight the corresponding changes in the manuscript; hence, I could not tell whether the authors revised the manuscript or not. I felt that both reviewers' comments are critical issues that readers may raise, so it is necessary for the authors to revise the manuscript accordingly. Regarding this revised manuscript, I still have the following major concerns about the reproducibility of results and the quality of the imputed values by ALRA.

In the previously submitted version, we included a version with changes highlighted. We feel it is unfortunate that the reviewer did not see this file, as it demonstrates the large number of improvements made to the manuscript in response to the comments made by all the reviewers. For this resubmission, we have attached a version which highlights changes from this round and the previous one, since these were missed by the reviewer.

1. In Figure 3A, the authors compared ALRA with several other imputation methods. However, the authors did not list the parameters used for the other imputation methods, and neither did they release the code for reproducing the results (the GitHub code available at <https://github.com/KlugerLab/ALRA-paper> was from three years ago and outdated). The code must be up-to-date so that all the results in the paper are reproducible.

All details relating to the the benchmarks reported in the paper are provided in the Online Methods section of the manuscript. In particular, when using other imputation methods, we used the default parameters. Regarding the code, we added a branch to our GitHub repository called "revisions" which contains the updated code. The following hypertext links directly to that branch: <https://github.com/KlugerLab/ALRA-paper>.

2. The authors showed that ALRA is better at preserving biological zeros in a dataset with only four cell types (Figure 2). In another analysis, they showed that ALRA is better at separating cell types when there are diverse cell types (Figure 3). The two conclusions are somehow contradictory: does ALRA work better with fewer cell types or more cell types? The authors need to make comparisons using the same metrics under the two settings (few cell types vs. multiple cell types).

In general, there is no contradiction. We showed that ALRA improves separation of cell types in six different datasets: Hrvatin et al., Tasic et al., Zeisel et al., Baron et al., Chen et al., and La Manno et al.. Some of these have many cell types and cell subtypes (e.g. Hrvatin et al. with 36 cell types/subtypes), whereas others have few (Baron et al. with 7). Thus, it is not clear why the reviewer suggests that ALRA is better at separating diverse cell types: in all of these settings, ALRA improves the separation compared to the original data.

As noted in the previous response, the numbers of zeros imputed may be affected by the number of different cell types. Suppose we have two datasets, A and B, which have different types of cells in each. In general, ALRA will preserve more biological zeros when A and B are imputed separately in contrast to when applied to the union of the two sets, $A \cup B$. The reason is that on the enlarged dataset, the negative zeros at each gene will in general be larger and hence more entries will be imputed to zero.

Finally, in practice users collect data for which the number of cell types and their diversity are often a priori unknown. In our manuscript, we have chosen a broad set of single-cell RNA-seq datasets, representative of how practitioners will use ALRA in their own analyses. Our manuscript showcases the ability of ALRA to preserve biological zeros and improve separation of cell types under a wide range of scenarios.

3. The distributions the authors plotted in Fig. S20 / S21 are worrisome, suggesting that ALRA cannot preserve the dynamic range of non-zero expression values before imputation. For example, among the three plots in the fourth row in Fig. S21, the leftmost plot shows a range of 1-2, but after imputation, the range expands to 1-5. Can authors justify these results?

In the previous submission, the leftmost plots showed the raw data, whereas ALRA takes the normalized data as input. In this revision, we modified the leftmost plots to be the normalized data. The dynamic range is now demonstrated to be nearly identical pre- and post-ALRA. We thank the reviewer for making this excellent point.

As an aside, note that the dynamic range may change slightly after imputation. Consider an extreme case, where two genes are highly correlated, but one is dramatically undersampled. After imputation, the maximum value of the undersampled gene may increase, as the largest values may not have been sampled in the original data. This can explain the small changes in dynamic range that are now reported in the updated Figures S20-21.

4. The correlations between the non-zero values before and imputation are very low: 0.23 - 0.56. This low correlation may indicate that ALRA's model assumption is not valid. Also, given that the non-zero expression values are altered greatly by ALRA, how should users interpret or use the imputed expression values output by ALRA?

The correlations of 0.23-0.56 were added following the reviewer's request in the previous round of review. We emphasize that correlation is not a good measure for accuracy or validity of ALRA. Let us illustrate this with two extreme examples: Suppose method "A" does nothing to the observed entries. In this case, the correlation between non-zero values before and after imputation is 1. In contrast, consider a method "B" that replaces each non-zero method with its true, underlying value (i.e. each gene's actual count of RNA molecules corresponding to each cell). Because of noise in the observed data, the correlation for this "oracle" method B would be considerably lower than 1.

ALRA denoises the non-zero values, bringing observed values closer to their true values. This is evidenced by improved performance of downstream analyses when compared to the observed measurements in 12 different datasets. Thus, we argue that the "low correlations" with the raw data are likely due to successful denoising of the data.

5. *Regarding the low-rank assumption, can the authors show some of the “gene modules” they have described in the response? For example, does a gene module (genes with large weights in a vector v_i) correspond to a set of genes that are highly correlated in a certain cell type? Why is this gene module biologically meaningful?*

We would like to clarify that this question is in fact not directly related to ALRA, but rather to the use of the leading eigenvectors of large matrices in biological data analysis, in particular in scRNA-seq studies.

Indeed, the low-rank assumption is ubiquitous in the analysis of genomics data and scRNA-seq in general. It is inherent in the utilization of principal component analysis (PCA) to reduce dimensionality of data, which is a key step in essentially every published analysis of scRNA-seq datasets. PCA represents the data using the top k principal components, for some number k . This is equivalent to taking a rank- k approximation to the data, as we do with ALRA. The only difference is that PCA typically requires centering of the columns (genes) before computing the singular value decomposition (SVD), whereas we apply SVD to the uncentered data.

This connection with PCA is a strength of ALRA. The use of ALRA’s imputed values as input to downstream analysis is essentially equivalent to using the PCs for downstream analysis, as is standard. In contrast to other methods that impute with complex statistical models, ALRA simply “imputes with PCA.” In fact, the additional step of restoring biological zeros by thresholding makes ALRA even more conservative than PCA.

In our previous response, we referenced the idea of “gene modules” as an intuitive explanation of the low-rank assumption. A gene module is a set of genes that are correlated in the dataset. In some cases, these may be controlled by the same transcription factor. In others, they may simply be markers of a cell type (i.e. they always co-appear in the same types of cells). For a dataset to be low-rank, it is only necessary for many variables (genes) to be correlated.

For example, consider the tutorial for Seurat, a widely popular software package for single cell data analysis. Under the section, “Perform linear dimensional reduction,” the authors show which genes loaded most strongly (positively or negatively) on the first five principal components. For example, the positively correlated genes CD79A, MS4A1, and CD79B are all markers of B cells, and these load strongly on the second principal component. These correlations result in a low-rank structure, which we take advantage of to impute technical zeros in the dataset.

REVIEWERS' COMMENTS

Reviewer #2 (Remarks to the Author):

The authors have successfully addressed my comments.